# River incision, $^{10}$Be production and transport in a source-to-sink sediment system (Var catchment, SW Alps)

Carole Petit[1], Tristan Salles[2], Vincent Godard[3,4], Yann Rolland[5,6], and Laurence Audin[6]

[1]Université Côte d'Azur, CNRS, Observatoire de la Côte d'Azur, IRD, Géoazur, 250 rue Albert Einstein, Sophia Antipolis 06560 Valbonne, France
[2]School of Geosciences, The University of Sydney, Sydney, NSW 2006, Australia
[3]Aix-Marseille Université, CNRS, Coll France, IRD, INRAE, CEREGE, Aix en Provence, France
[4]Institut Universitaire de France (IUF), Paris, France
[5]EDYTEM, Université Savoie Mont Blanc, CNRS, UMR 5204, Le Bourget du Lac, France
[6]ISTerre, Université Grenoble Alpes, Univ. Savoie Mont Blanc, CNRS, IRD, IFSTTAR, 38000 Grenoble, France

**Correspondence:** Carole Petit (carole.petit@univ-cotedazur.fr)

**Abstract.** Detrital $^{10}$Be from continental river sands or submarine sediments has been extensively used to determine the average long-term denudation rates of terrestrial catchments, based on the assumption that the rate of cosmonuclide production by interaction of source rocks with cosmic radiations balances out the loss of these nuclides by surface denudation. However, the $^{10}$Be signal recorded in sediments may be affected at the source by the response time of mountainous catchments to high-frequency forcings. In addition, transient sediment storage in piedmonts, alluvial plains, lakes or near the coast may also induce a difference between the erosive signal and its record in the sedimentary sink. Consequently, a significant part of the signal recorded in shallow-water sediments can be lost, as deep marine sediments may record simultaneously a signal coming from newly eroded source rocks along with one coming from the destabilization of previously deposited sediments.

In this paper, we use the landscape evolution model Badlands to simulate erosion, deposition and detrital $^{10}$Be transfer from a source-to-sink sedimentary system (the Var River catchment, Southern French Alps) over the last 100 kyr. We first compare model-based denudation rates with the ones that would be extracted from the $^{10}$Be record of local continental sediments (equivalent to river sands) and from off-shore deposited sediments over time, in order to examine if this record provides an accurate estimate of continental denudation rates. Then, we examine which conditions (precipitation rate, flexure, ice cover) permit to satisfy published measured river incision rates and $^{10}$Be concentration in submarine sediments.

Our results, based on the Var catchment cosmic ray exposure dating and modelling indicate that, while river sands do accurately estimate the average denudation rate of continental catchments, it is much less the case for deep submarine sediments. We find that deep sea sediments have a different, and often much smoother $^{10}$Be signature than continental ones, and record a significant time lag with respect to imposed precipitation rate changes, representing the geomorphological response of the margin. A model which allows us to fit both measured $^{10}$Be concentration in marine sediments and river incision rates on-land involves an increase in precipitation rates from 0.3 to 0.7 $\mathrm{m.yr^{-1}}$ after 20 ka, suggesting more intense precipitations starting at the end of the Last Glacial Maximum.

# 1 Introduction

Sedimentary deposits are important archives of the tectonic and climatic history of continents: for instance, the geometry, grain size, mineralogy and geochemical signature of deposits are impacted by changes in environmental conditions (e.g., relative sea level changes and precipitations over geologic times, as well as human activities during the Anthropocene (Syvitski et al., 2022). Provided good enough estimates of the transfer function between these sedimentary records and their external forcing, they can be reliable tools for reconstructing climatic cycles, subsidence curves, or monsoon onset for instance (Bentley et al., 2016; Li et al., 2016; Liu et al., 2016; Wan et al., 2006). However, depending on the considered timescales, the signatures in submarine sediments of some of these external (i.e., climatic or tectonic) forcings affecting aerial catchments depend on a myriad of processes which still remain difficult to extract from the deep sea record. To this end, one would need to evaluate not only how the eroded source responds to specific forcing but also how long and where are temporarily stored detrital sediments, and when are they re-injected into the system and eventually reach their sink.

Concerning the sediment source, mountainous catchments may not be very sensitive to high-frequency forcing, and the response time of these catchments may already affect the signal recorded in locally-produced sediments (e.g. Armitage et al., 2013; Godard and Tucker, 2021; Goren, 2016; Jerolmack and Paola, 2010). Second, transient sediment storage in piedmonts, alluvial plains, lakes or near the coast may induce a large time lag between the external signal and its record in the sedimentary sink (e.g. Blöthe and Korup, 2013; Clift and Giosan, 2014; Malatesta et al., 2018; Phillips and Slattery, 2006; Romans et al., 2016). Depending on considered timescales, the erosive signal itself can be completely buffered by this process (see a complete review in Romans et al. (2016)). Finally, submarine sediments can be reworked by gravitational processes, especially during sea-level falls (Phillips and Slattery, 2006). As a consequence, a significant part of the signal recorded in shallow-water sediments can be lost, whereas deep marine sediments may record simultaneously a signal coming from newly eroded source rocks and another one coming from the destabilization of previously deposited sediments. In addition, relative sea level variations may affect the connectivity between aerial rivers and submarine canyons, therefore limiting from time to time the efficiency of sediment transport in the offshore domain (Fryirs et al., 2007).

Detrital terrestrial cosmogenic nuclides (TCN, mostly [10]Be) concentrations from continental river sands or submarine sediments have been extensively used to determine the average long-term denudation rates of aerial catchments, provided enough quartz-bearing rocks outcrop at the surface to give a representative sampling of the whole catchment denudation (e.g. Bierman and Steig, 1996; von Blanckenburg, 2005; Lupker et al., 2012; Mandal et al., 2015; Siame et al., 2011; Vanacker et al., 2007). Denudation rate estimates from [10]Be concentration in quartz-rich sediments are often based on the assumption that the rate of TCN production by interaction of source rocks with cosmic radiations balances out the loss of these elements by surface denudation (Lal, 1991). Denudation rates can vary in time and space, which questions this steady-state assumption and may lead to under- or over-estimates of the true denudation rates (Bierman and Steig, 1996). The abundance of the target mineral (i.e., quartz in the case of [10]Be) in surface rocks may also vary, and has to be taken into account in order to correctly estimate the total production rate of a given catchment (Bierman and Steig, 1996; Safran et al., 2005; Carretier et al., 2015).

Moreover, sediments can be seen as an amalgamation of individual grains of different sizes and with different histories: allu-

vial terraces contain various clasts which TCN concentration is linked to the erosion rate of their catchment (inheritance) and to posterior TCN production (Repka et al., 1997). The distribution of TCN concentration in individual grains depend on the geomorphic processes acting at the source, and on the post-erosion TCN production (Codilean et al., 2010). Numerical models of individual grains journey have shown that some grains may have very long residence times in the piedmont (i.e., of the order of 100 ka) and may therefore be exported to their final depositional area long after they have been produced by bedrock erosion (Carretier et al., 2020). Landslides also may significantly affect denudation rate estimates from TCN if the catchment area is small (Yanites et al., 2009). Hence, understanding how detrital $^{10}$Be concentrations recorded in submarine sedimentological archives reflect denudation rates at the time of their deposition requires to quantify: i) how, how fast and where $^{10}$Be is produced; ii) how $^{10}$Be concentration in produced sediments is representative of average catchment denudation rates at any given spatiotemporal scale (Zerathe et al., 2022) and iii) how long it has taken for sediments, once they are produced, to reach the sedimentary sink where they have potentially been sampled.

In this paper, we adapt the surface process model Badlands (Salles, 2016) to simulate erosion, deposition and detrital $^{10}$Be transfer from a source-to-sink sediment system (the Var River catchment, Southern French Alps, and its marine depositional system in the Mediterranean sea) over the last 100 ka. We first compare simulated denudation rates with the ones that would be inferred from the $^{10}$Be concentration of continental sediments (equivalent to river sands) or off-shore deposited sediments at each time step, in order to assess at which timescales the steady-state assumption is valid, and if $^{10}$Be record in detrital sediments provides an accurate estimate of continental denudation rates. Then, we examine which conditions (precipitation rate, flexure, ice cover) permit to satisfy published river incision rates (Cardinal et al., 2022; Petit et al., 2019; Rolland et al., 2017, 2020; Saillard et al., 2014) and $^{10}$Be concentration in marine sediments (named hereafter $^{10}$Be$_{MS}$) in this particular, small-scale source-to-sink system (Mariotti et al., 2021).

## 2 Geomorphological and Geological setting

The Var catchment in the Southern French Alps is ideally suited to constrain source-to-sink processes: it is a relatively small catchment (~2800 $\mathrm{km}^2$), which encompasses some of the high altitude (~3000 m) summits of the Alpine Mercantour crystalline massif (Figure 1). The Var River has three main large tributaries: the Tinée and Vésubie Rivers, the headwaters of which are in the crystalline massif and the Esteron River, which flows only across the overlying meso-cenozoic sedimentary sequence. The hydrologic regime of the Var River, dominated by flash floods, is responsible for frequent hyperpycnal flows in the submarine domain (Mulder et al., 1998). The continental floodplain and shelf at the mouth of the Var River are very narrow, and most detrital sediments are deposited offshore at the foot of the Ligurian margin (Mediterranean Sea), at depths below -2000 m. The average annual discharge at the mouth of the Var river is ~50 $\mathrm{m}^3.\mathrm{s}^{-1}$, but it can reach more than 500 $\mathrm{m}^3.\mathrm{s}^{-1}$ during floods (Mulder et al., 1998). According to Mulder et al. (1998) and Syvitski et al. (2000), the suspended sediment load $C_s$ can be obtained from the following expression:

$$C_s = aQ_w^b \tag{1}$$

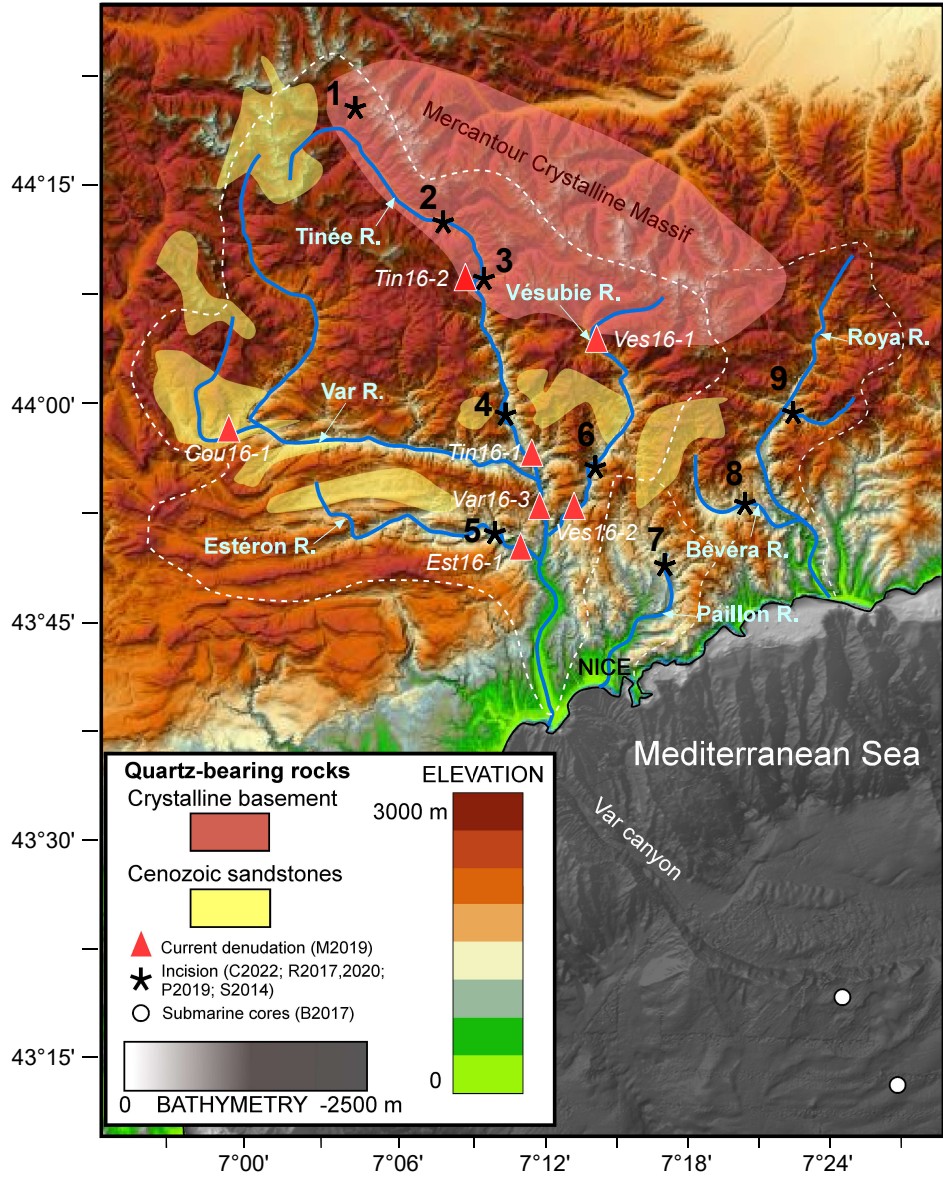

**Figure 1.** Topographic/bathymetric map and schematic outline of main quartz-bearing rock outcrops of the study area (transparent red and yellow patches). White dashed lines indicate the contours of the Var, Paillon and Roya Rivers catchments. Stars, open dots and triangles indicate the location of data constraints provided by surface exposure dating of river polished surfaces, after Saillard et al., 2014 (S2014), Rolland et al., 2017, 2020 (R2017,2020), Petit et al., 2019 (P2019), Cardinal et al., 2022 (C2022), geochemical analyses of submarine sediments after Bonneau et al., 2017 (B2017), and detrital [10]Be in river sands after Mariotti et al., 2019 (M2019), respectively. For river polished surfaces, the numbers refer to the following sites: 1 = Salso Moreno; 2 = Isola; 3 = Saint Sauveur; 4 = Lower Tinée; 5 = Estéron; 6 = Vésubie; 7 = Paillon; 8 = Bévéra; 9 = Roya. Italic letters refer to sample names in Mariotti et al. (2019).

where parameters $a$ $[M.L^{-3}][T^b.L^{-3b}]$ and $b$ relate the sediment concentration $C_s$ $[M.L^{-3}]$ to mean discharge $Q_w$ $[L^3.T^{-1}]$ at the river mouth, and have been estimated to ~7.7x10$^{-4}$ and ~1.65 for the Var River, respectively, from direct measurements of suspended sediment concentration and water discharge (Mulder et al., 1998). Modern average denudation rates estimates of the Var River catchment from [10]Be measurements in fluvial sediments range between 0.1 and 0.8 mm.yr$^{-1}$ (Mariotti et al., 2019). Two other coastal rivers flow East of the Var catchment: the Paillon and Roya Rivers, with much smaller drainage areas (258 and 601 km$^2$, respectively).

A previously published paper presented detailed sedimentological and geochemical analyses of sediment cores in the sedimentary ridge located at the outlet of the Var submarine canyon (Bonneau et al., 2016) (Figure 1). These analyses revealed a larger frequency of turbidite flows and slightly larger Epsilon-Nd ($\varepsilon_{Nd}$) values during the Last Glacial Maximum (LGM), which is interpreted as reflecting more intense erosion, especially in the crystalline massif, and larger sediment production during glacial periods. In these cores, [10]Be$_{MS}$ varies between 2x10$^4$ and 7x10$^4$ at.g$^{-1}$ (atoms per gram of quartz), which corresponds to average denudation rates of 0.2 to 0.5 mm.yr$^{-1}$ between 70 and 4 ka (Mariotti et al., 2021). On land, Cosmic Ray Exposure (CRE) ages of polished river cliffs have revealed fast incision rates of ~0.5 to 2 mm.yr$^{-1}$ during the late Pleistocene in most sites of the Var catchment and in the Bévéra River (Cardinal et al., 2022; Petit et al., 2019; Rolland et al., 2017; Saillard et al., 2014). River gorges located at high altitudes in the Mercantour Massif (Sites 1 and 2 on Figure 1, red and brown dots on Figure 2) show very fast incision (up to 4 mm.yr$^{-1}$) starting after the Younger Dryas (YD), which can be ascribed to a transient response of formerly glaciated valleys. Most of the lower altitude river gorges (Sites 3, 4, 5, 6, 8 on Figure 1, orange, yellow and green dots on Figure 2) start to be incised around 20 ka (i.e., close to the LGM). Two other sites (7 and 9 on Figure 1, cyan and blue dots on Figure 2) show much lower incision rates (<0.5 mm.yr$^{-1}$) extending from 0 to 80 ka.

Previously published interpretations of these data have suggested that rivers with sources in previously glaciated areas (Tinée, Vésubie) incised faster during and after the LGM not only because of increasing precipitations but also because of the massive release of glacier meltwaters and stored sediments occurring at that time (e.g. Saillard et al., 2014; Rolland et al., 2020). More recently, Cardinal et al. (2022) have pointed out a complex response of river systems of the SW French Alps to deglaciations, depending on their connection with glaciated areas and on the presence of lithological knickpoints.

To summarize, river incision data suggest: 1) transient and rapid incision following the Younger Dryas in high altitude areas; 2) steady, fast incision rates of ~1 mm.yr$^{-1}$ since the last 15-20 ka in almost all other points; 3) lower incision rates of ~0.2 to 0.5 mm.yr$^{-1}$ in the Paillon and Roya Rivers, east of the Var catchment. In all catchments, it appears that data points in the last ~20 ka range along slopes that define larger incision rates (sometimes by 1 order of magnitude) than average catchment denudation rates estimated from detrital [10]Be in river sands or marine sediments (Mariotti et al., 2019, 2021). This is not necessarily contradictory, as incision is a local phenomenon compared to the average catchment surface denudation. In particular, gorges where fast incision occurs can typically induce a transient decoupling from the catchment baselevel for the surrounding hillslopes (Reinhardt et al., 2007). However, there is a fundamental difference in the interpretation of river incision rates and [10]Be$_{MS}$ from deep marine sediments: while gorges bedrock surface exposure ages in the last 20 ka define high incision rates, suggesting that the post-LGM period was characterized by enhanced incision and gorges entrenchment, the geochemical sig-

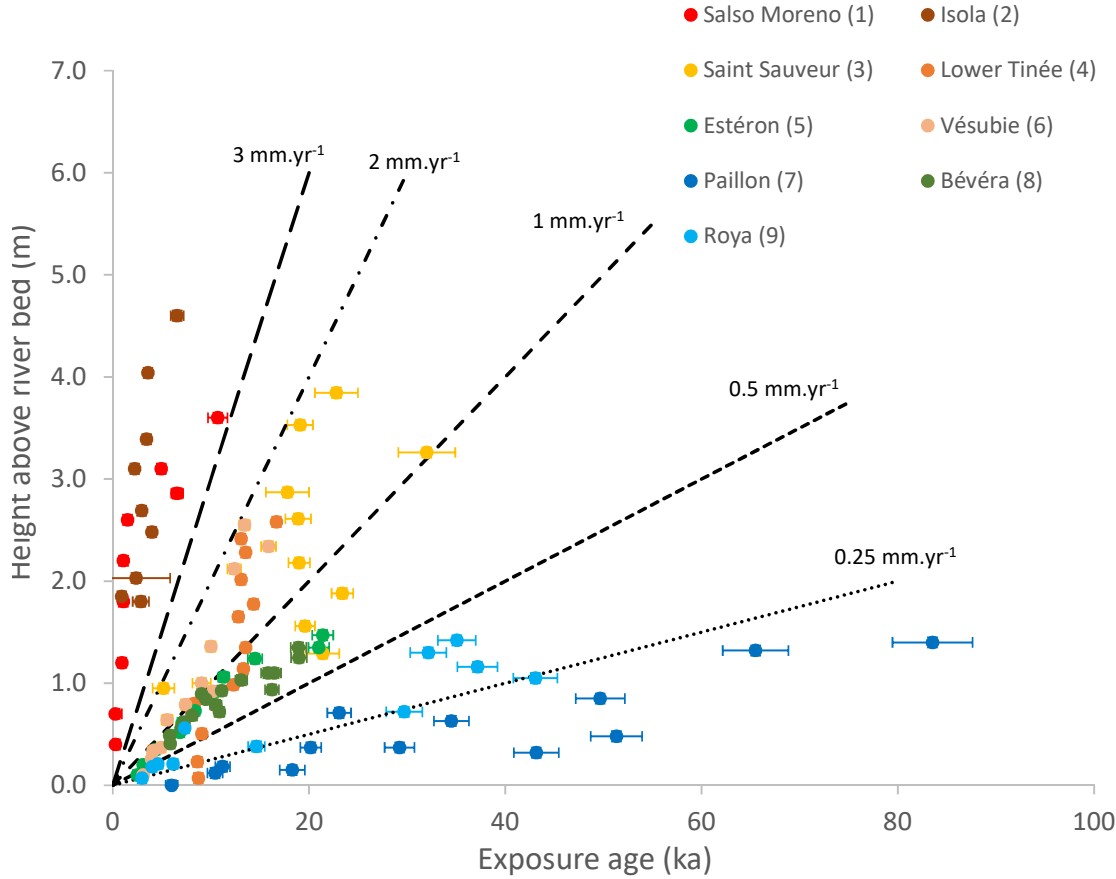

**Figure 2.** Altitude above the river bed vs. Cosmic Ray Exposure (CRE) ages and corresponding error bars in samples of river-polished surfaces of the Var, Roya and Paillon catchments (Cardinal et al., 2022; Rolland et al., 2020, 2017; Petit et al., 2019; Saillard et al., 2014). See location of the sites on Figure 1. Dashed lines indicate constant incision rate curves, for comparison.

nature of marine sediments is interpreted (Bonneau et al., 2016; Mariotti et al., 2021) as reflecting more intense erosion during the glacial episodes than during deglaciations.

## 3 Methods

### 3.1 Landscape evolution model

We use the Landscape Evolution Model (LEM) Badlands (Salles, 2016) to simulate erosion and deposition on an irregular TIN topographic grid under various tectonic (uplift) and climatic (precipitation, hillslope processes) time and space parameters. River incision is simulated using the Stream Power Law (Whipple and Tucker, 1999):

$$\varepsilon = K(\bar{P}.A)^m S^n \tag{2}$$

where $\varepsilon$ is the erosion rate $[L.T^{-1}]$, $K$ is a dimensional coefficient that describes the erosional efficiency $[L^{1-2m}.T^{-1}]$, $A$ is the drainage area $[L^2]$ and $\bar{P}$ is the spatial and temporal variation in precipitation rate $P$ relative to a mean precipitation rate $P_0$ (1 m.yr$^{-1}$), the product being used as a proxy for the discharge, $S$ is the channel slope, $m$ and $n$ are positive exponents. A $m/n$ value close to 0.5 was previously estimated in the Var and Vésubie catchments (Saillard et al., 2014; Petit et al., 2017), so we chose to set in all models $m$ and $n$ to classically used values of 0.5 and 1, respectively, and to tune the $K$ value in order to fit measured incision rates. Hillslope processes on land and short-distance sediment transport at sea can be simulated by a linear diffusion law:

$$\frac{dh}{dt} = K_D \nabla^2 h \tag{3}$$

where *dh/dt* is the altitude change due to diffusive processes $[L.T^{-1}]$, $h$ is the altitude $[L]$ and $K_D$ is the hillslope diffusion coefficient $[L^2.T^{-1}]$, which can vary between the continental and marine domains. We fix the diffusion coefficient to low values (0 to 0.025 m$^2$.yr$^{-1}$) both on land and at sea in order to ensure that any observed smoothing effect on the [10]Be record in deposited sediments is not due to diffusive processes (Table 2). For the same reason, we do not consider non-linear diffusion components. For river systems, we use the detachment-limited law (Eq. 2) but we impose sediment deposition either when the channel slope falls below a given threshold (alluvial plain deposition, see Table 3) or when the rivers reach their baselevel or an endorheic depression. In these cases, available sediment fluxes carried by rivers are used to compute the volume of sediments to deposit. If transported sediment fluxes, when deposited, are insufficient to fill the depression or to reach the prescribed channel slope threshold, all sediments would be deposited and the outgoing river sediment concentration would be null. If, on the other hand, the available sediment flux exceeds the required deposition volume, the excess flux will be carried out to the downstream nodes.. A low critical slope of 0.5% is applied, except for two models where a very low threshold of 0.01 % is used in order to drastically limit alluvial plain deposition. In addition, we take into account submarine sediment transport in order to simulate the occurrence of hyperpycnal flows. Following an approach similar to Petit et al. (2015) and Thran et al. (2020), we assume that hyperpycnal flows occur when the sediment load at the river mouth is larger than a given threshold. If the flow density exceeds this threshold, instead of being deposited near the baselevel, sediments continue their route along the submarine slope.

In addition, we assume that the flow does not incorporate water along its path. The flow density at the river mouth $\rho_f$ [$M.L^{-3}$] can be computed: i) either with a mass estimate from water and sediment discharge ($Q_w$ and $Q_s$, respectively [$L^3.T^{-1}$]) and densities ($\rho_w$ ans $\rho_s$, respectively):

$$\rho_f = \frac{\rho_w.Q_w + \rho_s.Q_s}{Q_w + Q_s} \tag{4}$$

or using the rating parameters $a$ and $b$ (Syvitski et al., 2000), which can be determined for each river system from discharge and sediment load measurements.While Eq. 4 considers the total mass of sediment transported by the river (i.e., bedload and suspended load), the rating parameters allow an empirical estimation of the suspended load, the one that effectively contributes to the increase in flow density. We consider that submarine flow can trigger bedrock erosion as for aerial channels, but the parameters of the stream power law are adjusted in order to account for: 1) constant drainage area along channel length in the submarine domain and 2) lower shear stress on the submarine channel bed compared to aerial rivers, which is simplified assuming an effective slope $S_{eff}$ such that:

$$S_{eff} = S\frac{\rho_s - \rho_w}{\rho_s} \tag{5}$$

Deposition occurs in the submarine domain either for gentle slopes (similar to alluvial plains on land) and beneath a certain depth (-2300 m in most models) corresponding to the depth of the abyssal plain. Flexural isostasy can be incorporated with a constant or space-variable effective elastic thickness (*EET*) used to compute the vertical motion resulting from the response of the lithosphere to loading (by ice, sedimentation or sea level rise) or unloading (deglaciation, erosion or sea level drop). Flexural isostatic response of the lithosphere is computed using the flexure equation:

$$D\nabla^4 w(x,y) + \Delta\rho g w(x,y) = L(x,y) \tag{6}$$

Where $w$ is the vertical deflection, $\Delta\rho$ is the density contrast between the mantle and the filling material, $L$ is the load (N) and $D$ is the flexural rigidity of the lithosphere (N.m):

$$D = E.EET^3/12(1-\nu^2) \tag{7}$$

With $E$ the Young's modulus and $\nu$ the Poisson's ratio, equal to $10^{11}$ Pa and 0.25, respectively. The flexure module in Badlands uses the gFlex package (Wickert, 2016). Apart from flexural response to erosion/sedimentation and ice and sea water loading and unloading during the model run, no vertical motions are applied to the topography.

## 3.2 Ice cover and sea level changes

As the Mercantour massif was periodically covered by glaciers during the Quaternary, we simulate the ice thickness and extent at every time step assuming that the LGM corresponds to the maximum ice extent map (Brisset et al., 2015). We consider that ice thickness varies with Mediterranean sea surface temperatures (SST), which ranged between ~5°C (LGM) and ~15°C during the considered time period (Hayes et al., 2015; Rodrigo-Gamiz et al., 2013). Glacial periods with full ice extent are imposed for SST lower than 6.5°C and complete deglaciation for SST above 11°C. Between these thresholds, the ice thickness

is assumed to vary linearly with the SST. In order to avoid rapid variations of the ice cover, the SST curve is smoothed using a 5 kyr sampling interval which is then resampled at 1 kyr step using cubic interpolation (Figure 3). When ice thaws, an equivalent amount of water (assuming a ratio between the ice and water heights of 0.93) is released as runoff. Sea level variations can be imposed according to the Mediterranean eustatic sea level curve published in Waelbroeck et al. (2002). Badlands does not simulate glacial erosion; however, we must consider a non-null erosion rate beneath glaciated areas in order to avoid over-estimation of the [10]Be concentration in sediments produced by basement erosion after glaciers retreat. For this purpose, we simulate the in-situ erosion and sediment production due to glacial processes by increasing the local hillslope diffusion coefficient proportionally to the ice thickness, which will locally enhance denudation beneath glaciated areas, while river discharge is set to zero. This simplified representation is based on the assumption that glacial sediments fluxes are proportional to the topographic slope (hence to ice velocity) and that glacial erosion is related to the shear stress exerted by the glacier on the bedrock (Boulton, 1996). As glacial erosion is simulated by local processes (diffusion) and river discharge is set to zero beneath glaciers, the export of glacial sediments is very low during glaciations. Besides erosion, the effect of ice coverage is twofold: it blocks cosmic radiations so [10]Be production is null beneath in areas covered by glaciers, and it creates a positive vertical load and downward flexure of the lithosphere.

## 3.3 [10]Be production and transport

[10]Be production rates by neutron spallation and muon capture are the same as in Mariotti et al. (2019) and computed according to Braucher et al. (2011); Martin et al. (2017) for a latitude of 40° (Table 1) using the scaling parameters by Stone (2000). Earth magnetic field variations are not considered in this study. The topographic shielding is computed from the TIN topographic grid. A shielding correction can be applied to account for the topographic smoothing due to the DEM resolution, which tends to underestimate the actual shielding (Norton and Vanacker, 2009).

A map of quartz-bearing rocks is defined according the geological map of Nice and its hinterland (Rouire et al., 1980); most quartz-bearing rocks correspond either to granitic and metamorphic Palaeozoic basement rocks in the Mercantour massif or to Cenozoic sandstones in the sedimentary cover (Figure 1). The average quartz concentration in source rocks is fixed at 50%. The initial [10]Be concentration in quartz-bearing rocks is computed assuming a steady-state average denudation rate for the whole grid. [10]Be concentration $N(z,t)$ varies with time and depth, and we simply compute it at the surface ($z$=0) of eroded domains (Lal, 1991):

$$\frac{dN(0,t)}{dt} = P(0,t) - \left(\lambda + \frac{\rho.\varepsilon(t)}{\Lambda}\right) N(0,t) \tag{8}$$

where $N$ is the [10]Be concentration (at.g$^{-1}$), $P$ is the production rate (at.g$^{-1}$.yr$^{-1}$), $\lambda$ is the [10]Be radioactive decay constant (yr$^{-1}$), $\rho$ the rock density (g.cm$^{-3}$), $\Lambda$ is the attenuation length (g.cm$^{-2}$), and $\varepsilon$ the erosion rate (cm.yr$^{-1}$). At each time step, the production rate is computed taking into account the quartz abundance of the source rock and the potential shielding of cosmic rays by the surrounding topography, and/or by the ice or sea cover.

[10]Be production results primarily from neutron spallation, and fast and slow muon capture with different production rates and attenuation lengths (Braucher et al., 2011) (Table 1). Assuming that erosion and production rates are constant during a given

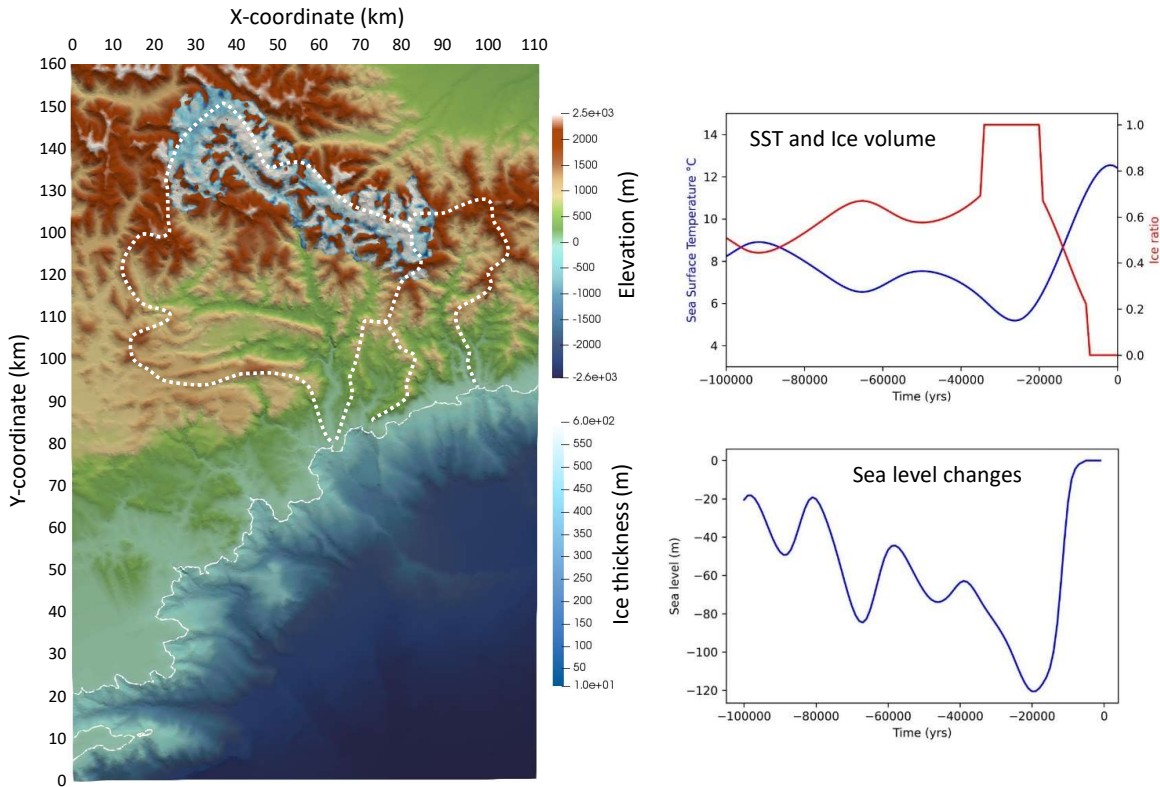

**Figure 3.** Left : Topography, bathymetry and maximum ice thickness (after Brisset et al., 2015), and contour of the main catchments (dashed line); Top right: smoothed variations of the sea surface temperature (SST, after Hayes et al., 2005 and Rodrigo-Gamiz et al., 2013) in blue and ice thickness ratio (1 is for full ice, 0 for no ice) in red; bottom right: sea level variations (after Waelbroeck et al., 2002).

time step of the model, we can compute explicitly the [10]Be concentration in each eroded node of the source rocks at each time step, without any a priori steady-state assumption (Knudsen et al., 2019):

$$N(0,t) = N(0,t-\Delta t).\exp\left[-\left(\lambda + \frac{\rho.\varepsilon(t)}{\Lambda}\right)\Delta t\right] + \frac{P(0,t)}{\lambda + \frac{\rho\varepsilon(t)}{\Lambda}}.\left(1 - \exp\left[-\left(\lambda + \frac{\rho\varepsilon(t)}{\Lambda}\right)\Delta t\right]\right) \tag{9}$$

This Eulerian formulation, where the erosion rate ε is equal to the vertical velocity at which the rock material is vertically advected up to the surface is chosen because it does not necessitate to compute vertical concentration profiles for each grid point.

For the initial [10]Be concentration N(0,0) we assume steady-state between [10]Be production and erosion by imposing a mean long-term erosion rate $\varepsilon_0$ :

$$N(0,0) = \frac{P(0,0)}{\frac{\rho.\varepsilon_0}{\Lambda} + \lambda} \tag{10}$$

At each time step, the total sediment, quartz and [10]Be fluxes are computed on each grid node. In the case of deposited sediments, we then compute the mean detrital [10]Be concentration and quartz content of sediments, knowing the volume contribution, quartz proportion and [10]Be concentration of each eroded source to the total amount of deposited sediments, assuming a perfect mixing between all sources. We use as initial conditions a smoothed topographic and bathymetric DEM with a spatial resolution of 500 m x 500 m.

In a first test, a simulation is run with a constant precipitation rate of 0.5 m.yr$^{-1}$ over 5000 years with a timestep of 100 years in order to calibrate SPL parameters, mainly the erodibility coefficient *K* against the results of Mariotti et al. (2019). From this model (M1), we compute the [10]Be production rate in the Mercantour massif, the mean [10]Be concentration of continental river sediments corresponding to the catchments sampled by Mariotti et al. (2019) and the related mean catchment denudation rate deduced from these concentrations, with an initial steady-state [10]Be concentration computed using a mean denudation rate of 0.2 mm.yr$^{-1}$, in agreement with Mariotti et al. (2019) (Figure 4 and Table 2). Discrepancies between our model and the results of Mariotti et al. (2019) can arise from: 1) the fact that we use a smoother topographic grid and a simpler map of quartz-bearing areas (for [10]Be production rate mismatches) and 2) the fact that we compute erosion for each grid node, then average it over the catchment area, instead of considering a spatially homogeneous catchment denudation rate (for erosion rate mismatches). Data from the Coulomp River, a small tributary of the Var River, could not be satisfyingly reproduced (Table 2, Cou-16.1): the predicted low [10]Be concentration resulted in a denudation rate more than twice that of Mariotti et al. (2019). Similarly, one point in the Vesubie River (Ves-16.1), which has a large [10]Be production rate, shows a modelled [10]Be concentration slightly larger than measured by Mariotti et al. (2019). As a consequence, the erosion rate for this point is underestimated (Figure 4). While not perfect, simulated estimates of [10]Be production and erosion rates provide a good fit with observations and allow us to constrain the value of *K*. Several improvements might reduce the observed discrepancies, first by using a higher DEM resolution and second by accounting for underground (karstic) water circulation known to take place in the region like for the Coulomp River (Audra et al., 2009), which should modify river discharge and related incision rates.

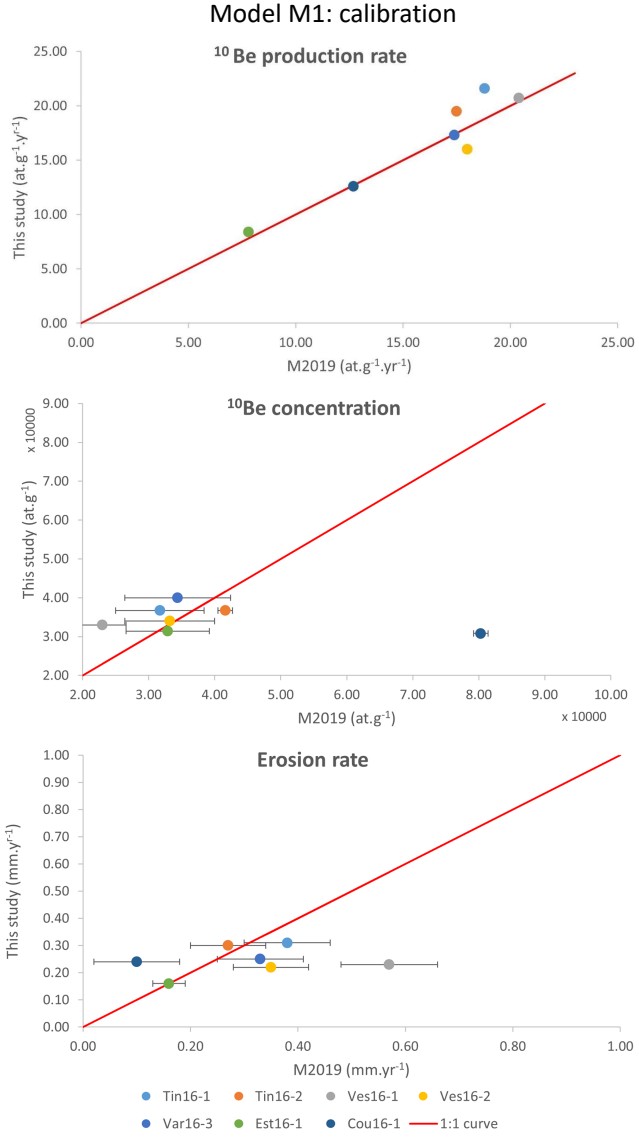

**Figure 4.** Comparison between the results of Mariotti et al. (2019) (x-axis) and the short-term model (this study, y-axis) for several rivers of the Var catchment, for [10]Be production rates (top), [10]Be concentration in river sediments (middle) and steady-state erosion rates (bottom). Red line indicates the 1:1 slope. See Table 2 for values.

**Table 1.** Sea level high latitude parameters for [10]Be production (after Braucher et al. (2011))

| Neutron spallation rate $at.g^{-1}.a^{-1}$ | Slow muon capture rate $at.g^{-1}.a^{-1}$ | Fast muon capture rate $at.g^{-1}.a^{-1}$ | Neutron attenuation length $g.cm^{-2}$ | Slow muon attenuation length $g.cm^{-2}$ | Fast muon attenuation length $g.cm^{-2}$ | Radioactive decay constant $a^{-1}$ | Density $g.cm^{-2}$ |
|---|---|---|---|---|---|---|---|
| 4.11±0.19 | 0.011±0.001 | 0.039±0.004 | 160 | 1500 | 4320 | 4.9867x10$^{-7}$ | 2.5 |

**Table 2.** Calibration of [10]Be production and erosion rates after the results of Mariotti et al. (2019) for the present-day rates

| Sample site | Production rate $at.g^{-1}.yr-1$ | | [10]Be concentration $at.g^{-1}$ | | Erosion rate $mm.yr^{-1}$ | |
|---|---|---|---|---|---|---|
| | M2019 | This study | M2019 | This study | M2019 | This study |
| Tin16-1 | 18.8 | 21.6 | (3.17±0.67)x10$^4$ | 3.67x10$^4$ | 0.38±0.08 | 0.31 |
| Tin16-2 | 17.5 | 19.5 | (4.16±0.11)x10$^4$ | 3.67x10$^4$ | 0.27±0.07 | 0.30 |
| Ves16-1 | 20.4 | 20.7 | (2.3±0.36)x10$^4$ | 3.30x10$^4$ | 0.57±0.09 | 0.23 |
| Ves16-2 | 18.0 | 16.0 | (3.32±0.68)x10$^4$ | 3.40x10$^4$ | 0.35±0.07 | 0.22 |
| Var16-3 | 17.4 | 17.3 | (3.44±0.80)x10$^4$ | 4.00x10$^4$ | 0.33±0.08 | 0.25 |
| Est16-1 | 7.8 | 8.4 | (3.29±0.63)x10$^4$ | 3.14x10$^4$ | 0.16±0.03 | 0.16 |
| Cou16-1 | 12.7 | 12.6 | (8.03±0.11)x10$^4$ | 3.08x10$^4$ | 0.10±0.08 | 0.24 |

## 4 Results

### 4.1 Record of time variable erosion rates in river sands and at sea

We then test the response of a topographic grid representing the Var aerial and submarine systems to climatic (precipitation) variations. The model runs for 100 kyr with an adaptive time step of max. 1000 years. We keep a constant rock erodibility of $5x10^{-6}$ yr$^{-1}$ and a hillslope diffusion coefficient of $2.5x10^{-2}$ m$^2$.yr$^{-1}$. The river sediment load is computed from water discharge using rating parameters (Syvitski et al., 2000) for the Var River *a* and *b* equal to $1x10^{-3}$ and 1.6, respectively, and a low threshold flow density (equal to water density), which insures that all river sediments are exported to the deep submarine basin and that no large river delta is formed near the coastline.

In a first series of tests, we want to investigate how well the [10]Be concentrations in continental (i.e., in river sand) and submarine (turbidite-like) deposits compare with the average catchment denudation rate directly output from the model. For this purpose, we select the total area of the Var catchment where [10]Be is produced and compute the average [10]Be concentration $N_{ex}$ in the volume of rock eroded from this source for each time step (hereafter called "in-situ" sediments, because they are not yet

**Table 3.** Model parameters

| Figure | Model | Precipitation $m.yr^{-1}$ | Erodibility $yr^{-1}$ | Diffusion coefficient $m^2.yr^{-1}$ | | Critical slope | Deposition depth ($m$) | Sea level changes | Ice cover | Flexure |
|---|---|---|---|---|---|---|---|---|---|---|
| | | | | aerial | marine | | | | | |
| 4 | M1 | 0.5 | $4.5 \times 10^{-6}$ | 0 | $2.5 \times 10^{-2}$ | 0.005 | -2300 | No | No | No |
| 6A | M2 | 0.25/1 | $5.0 \times 10^{-6}$ | 0 | $2.5 \times 10^{-2}$ | 0.005 | -2300 | No | No | No |
| 6B | M3 | 0.25/1 | $5.0 \times 10^{-6}$ | 0 | $2.5 \times 10^{-2}$ | 0.005 | -2300 | Yes | No | No |
| 6C | M4 | 0.25/1 | $5.0 \times 10^{-6}$ | 0 | $2.5 \times 10^{-2}$ | 0.005 | -2300 | Yes | Yes | No |
| 6D | M5 | 0.25/1 | $5.0 \times 10^{-6}$ | 0 | $2.5 \times 10^{-2}$ | 0.005 | -2300 | Yes | Yes | Yes |
| 6E | M6 | 0.25/1 | $5.0 \times 10^{-6}$ | 0 | $2.5 \times 10^{-2}$ | 0.0001 | -200 | No | No | No |
| 6F | M7 | 0.25/1 | $5.0 \times 10^{-6}$ | 0 | $2.5 \times 10^{-2}$ | 0.0001 | -800 | No | No | No |
| 7 | M8 | 0.5 | $5.0 \times 10^{-6}$ | 0 | $2.5 \times 10^{-2}$ | 0.005 | -2300 | Yes | Yes | Yes |
| 8-9A | M9 | 0.3/0.7 | $3.5 \times 10^{-6}$ | $1.0 \times 10^{-3}$ | $2.5 \times 10^{-2}$ | 0.005 | -2300 | Yes | Yes | Yes |
| 9B | M10 | 0.3/0.7 | $3.5 \times 10^{-6}$ | $1.0 \times 10^{-3}$ | $2.5 \times 10^{-2}$ | 0.005 | -2300 | Yes | Yes | No |
| 9C | M11 | 0.3/0.7 | $3.5 \times 10^{-6}$ | $1.0 \times 10^{-3}$ | $2.5 \times 10^{-2}$ | 0.005 | -2300 | Yes | No | Yes |
| 9D | M12 | 0.3/0.7 | $3.5 \times 10^{-6}$ | $1.0 \times 10^{-3}$ | $2.5 \times 10^{-2}$ | 0.005 | -2300 | No | Yes | Yes |
| 9E | M13 | 0.5 | $3.5 \times 10^{-6}$ | $1.0 \times 10^{-3}$ | $2.5 \times 10^{-2}$ | 0.005 | -2300 | Yes | Yes | Yes |
| 9F | M14 | 0.3/0.7 | $3.5 \times 10^{-6}$ | $1.0 \times 10^{-3}$ | $2.5 \times 10^{-2}$ | 0.005 | -2300 | Yes | No | No |
| 9G | M15 | 0.3/0.7 | $3.5 \times 10^{-6}$ | $1.0 \times 10^{-3}$ | $2.5 \times 10^{-2}$ | 0.005 | -2300 | No | Yes | No |

transported nor deposited), from the contribution of each catchment node i such as:

$$\overline{N_{ex}} = \frac{\sum_{i=1}^{n} N_i \varepsilon_i Q_i}{\sum_{i=1}^{n} \varepsilon_i Q_i} \tag{11}$$

where $Q_i$ is the quartz abundance, $N_i$ is the $^{10}$Be concentration ($at.g^{-1}$) and $\varepsilon_i$ the eroded mass of sediments (in g). Then we use Eq. 10 to compute the catchment denudation rate assuming a steady-state condition, and compare it with the average "actual" denudation rate directly output from the model (i.e., the average volume of eroded sediments in the same area per time step). Finally, we extract the average $^{10}$Be$_{MS}$ value in 8x8 km square areas located at the mouth of the Var River for model M6, in the submarine canyon for model M7 and in the deep submarine basin for models M2 to M5. We then use the same equation to compute the average catchment denudation rate as recorded by deep sea sediments (as shown in Figure 5). Simulations are run with alternating low (0.25 m.yr$^{-1}$) and high (1 m.yr$^{-1}$) precipitation rate periods lasting 20 kyr each (Figure 6). The first simulation (M2) is run with a constant sea level, no ice cover and no lithospheric flexure; then we successively implement a variable Mediterranean sea-level, ice cover in the Mercantour massif (M3 and M4, see Figure 3) and lithospheric flexure with a constant *EET* of 20 km (M5), which corresponds to a moderately rigid lithosphere where the crust and mantle elastic lids are decoupled (Burov and Diament, 1995). The last two models of this series (M6 and M7) have similar parameters as M2, except

that i) we drastically decrease the slope threshold for alluvial plain deposition, which almost impedes sediment storage on-land and ii) we force deposition in the submarine domain at shallower depths: either in the coastal delta (-200 m) for M6 or in the submarine canyon (-800 m) for M7.

The first model (M2, Figure 6A), where there are no sea level changes, no ice cover and no flexural isostatic response of the lithosphere shows that the two periods of more intense precipitations are well-detected in in-situ sediments, which [10]Be signature gives erosion rates consistent with the actual ones. Marine sediments also seem to record two periods of larger erosion rates, but the first one (between 25 and 45 ka) is barely visible and the peak of the second one (80-90 ka) occurs 10 to 20 ka after the middle of the second high-precipitation period. The second model (M3, Figure 6B) where sea level variations are

present displays approximately the same behaviour as M2 for in-situ sediments, with apparent erosion rates consistent with real ones. Erosion rates computed from marine sediments also show fluctuations consistent with two periods of more intense precipitations, but they appear much smoother and offset by 20 ka, compared to the actual ones. In the two following models where ice cover is present (M4 and M5, Figures 6C and 6D), the effect of partial coverage of the Mercantour massif by ice during almost all the duration of the model is twofold: i) it reduces the effect of precipitation variations, since no run-off occurs

beneath the ice cover and ii) it reduces the production and exportation of [10]Be-poor sediments, except during the last period (80-100ka) where the massif is free of ice. As a consequence, while in-situ sediments still record well the actual erosion rate, the [10]Be signal in the submarine domain strongly under-estimates it. Finally, models similar to M2 but with a lower critical slope and where sediment deposition is forced either in the delta (M6) or in the submarine canyon (M7) show that marine sediments of the delta record well the onset of first precipitation pulse, with only a small time lag (5 ka). Then, the apparent

erosion rate computed from marine sediments progressively decreases and the second period of intense precipitation is not recorded. In this model, [10]Be-poor sediments coming from the second precipitation pulse are not deposited in the sampled area but further south, due to delta progradation. Finally, if deposition is forced to occur in the middle of the submarine slope, the first precipitation peak is still detected but it appears smoother, of lower amplitude, and it occurs 20 ka later than on land (Figure 6F).

In simulations without ice cover (Figures 6A, 6B, 6E and 6F), denudation rates estimated from in-situ sediments do record the succession of periodic pulses but generally overestimate the actual denudation rate by a few percent. This overestimation reflects a more intense erosion in low-altitude river thalwegs than on the hillslopes and high-altitude interfluves, which slightly promotes the export of [10]Be-poor sediments with respect to [10]Be-rich ones, hence increasing the apparent denudation rate.

For all simulations, both in-situ and actual denudation rate evolutions depict more complex patterns than the imposed climatic

(precipitation) forcing. Short-period, low-amplitude variations are visible, which are related to local and internal adjustments of the modelled topography and not to the external forcing, although sea level variations seem to slightly modify their amplitude (Figures 6A and 6B). There is no clear time lag between the onset of higher precipitation periods and their record in in-situ sediments, but the apparent denudation rate from in-situ river sediments displays a curved shape at the transition between low and high precipitation rate periods (well visible for instance on figure 6A between 60 and 80 ka), which corresponds to

well-known analytical solutions for periodic changes in erosion rates (Bierman and Steig, 1996) and is consistent with the detachment-limited hypothesis. This reflects the time needed for [10]Be concentration to reach a steady-state relative to the

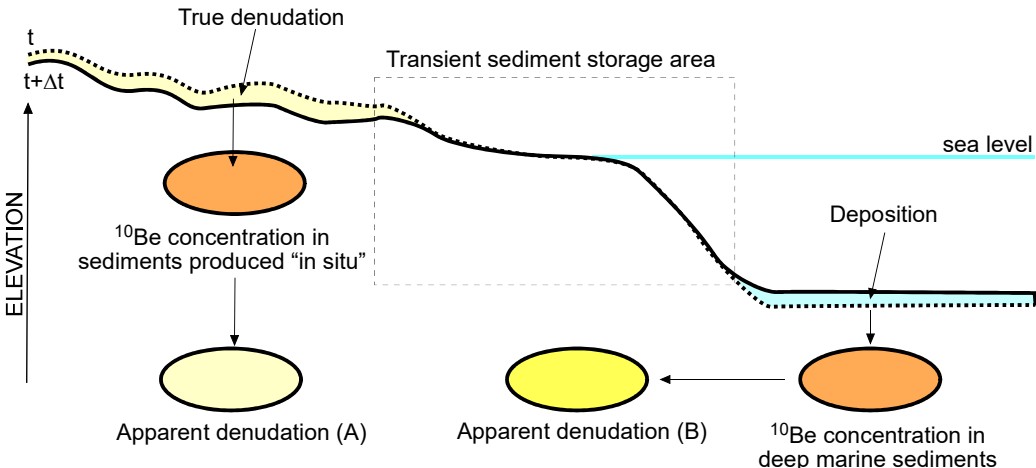

**Figure 5.** Schematic description of the approach used to compare the apparent denudation for a given time step computed from the mean [10]Be concentration of in-situ produced sediments (left part of the topographic profile, apparent denudation A) or from the mean [10]Be concentration of marine sediments ([10]Be$_{MS}$, right part of the profile, apparent denudation B) with the true denudation.

massif denudation rate (~5 ka); this effect is also visible at the transition from high to low precipitation rates (Figures 6E and 6F).

This series of tests seem to indicate that: i) the in-situ produced alluvial sediments record well the variations in the rate of denudation, although they may slightly overestimate them; ii) [10]Be of submarine sediments [10]Be$_{MS}$ in the deepest part of the basin does not allow to retrieve sharp climate variations especially if glaciers cover the area where [10]Be is produced for a long period of time, but correctly estimate the average long-term erosion rate, with only smooth variations; iii) finally, there can be a significant time lag between the middle of the high precipitation periods and the peak in denudation rates recorded in submarine sediments.

In order to refine our understanding of the smoothing and time lag effects, we aimed at tracing artificially-enriched [10]Be-rich sediments in the mountain range down to the submarine basin. To do so, we imposed a constant and artificial, large [10]Be concentration in high-altitude areas (above 1800 m) over a 5 kyr period from 30 to 35 ka. Then, we compare the [10]Be signature of deep-sea sediments between two identical models, one ran with the [10]Be enrichment and one without (model M8, Figure 7). We do not show all the tests here for the sake of simplicity, but rather illustrate a typical signature of this transient [10]Be enrichment, as visible in submarine sediments. This result is obtained accounting for a variable sea level, ice cover and lithospheric flexure, and a constant precipitation rate of 0.5 m.yr$^{-1}$ (Table 3). If we consider the earliest arrival of [10]Be-rich sediments in the submarine basin, the time lag from source to sink appears to be relatively small (1-3 kyr) since [10]Be-rich sediments arrive in the basin shortly after they begin to be produced in the massif (Figure 7); however, the time lag between

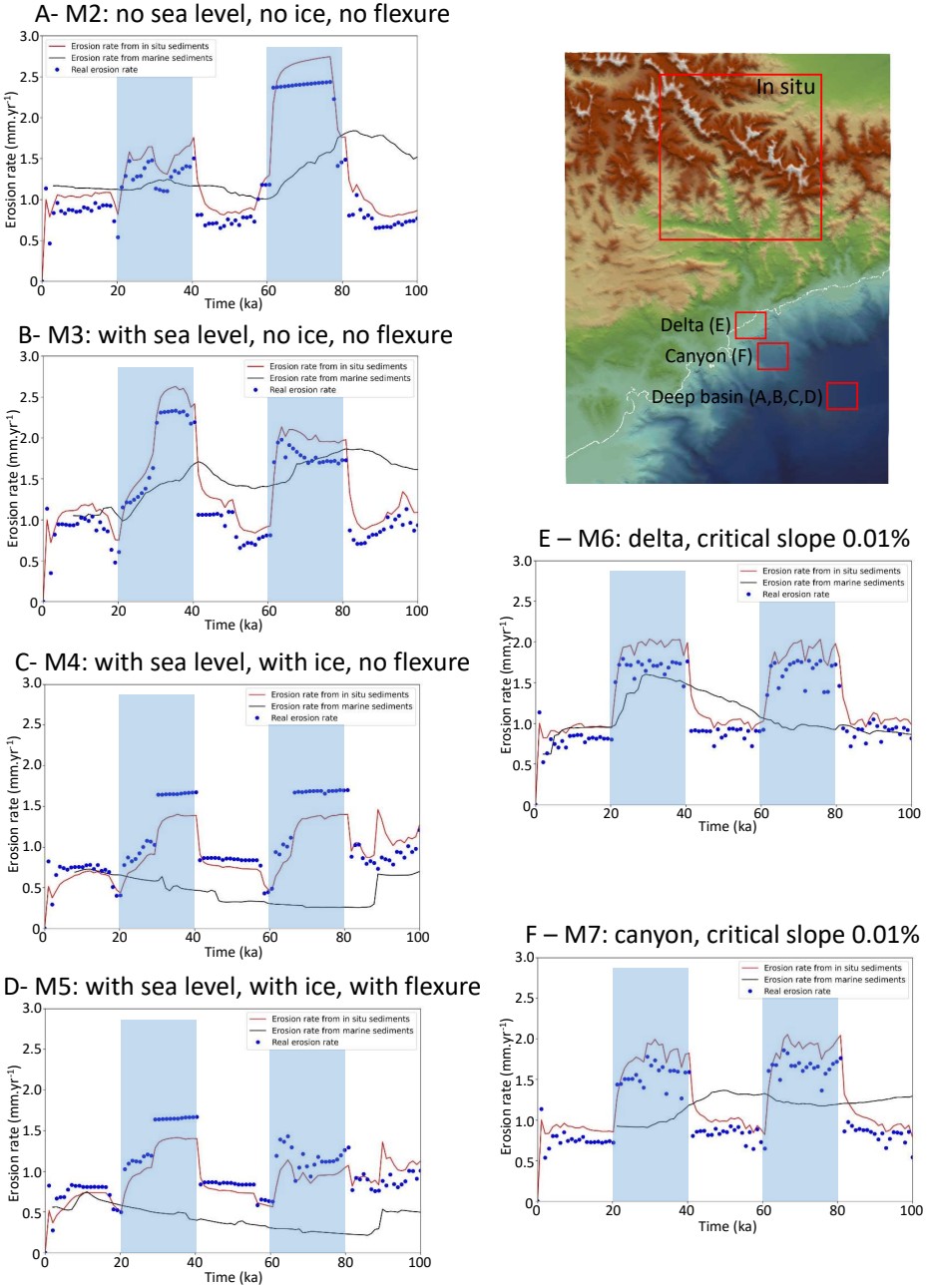

**Figure 6.** Test of the apparent denudation rates given by the [10]Be concentration of in-situ (red line) and deep marine ([10]Be$_{MS}$, black line) sediments, versus the actual one extracted from the model (blue dots). Periods of large precipitation rates are indicated by the transparent blue rectangles. A: model with only variable precipitation rate; B: model with variable precipitation rate and sea level variations; C: model with variable precipitation rate, ice cover and sea level variations; D: model with variable precipitation rate, sea level variations, ice cover and flexure; E: model similar to A but with a lower critical slope for sediment deposition, and forced deposition below -200m; F: model similar to E but with forced deposition below -800m.

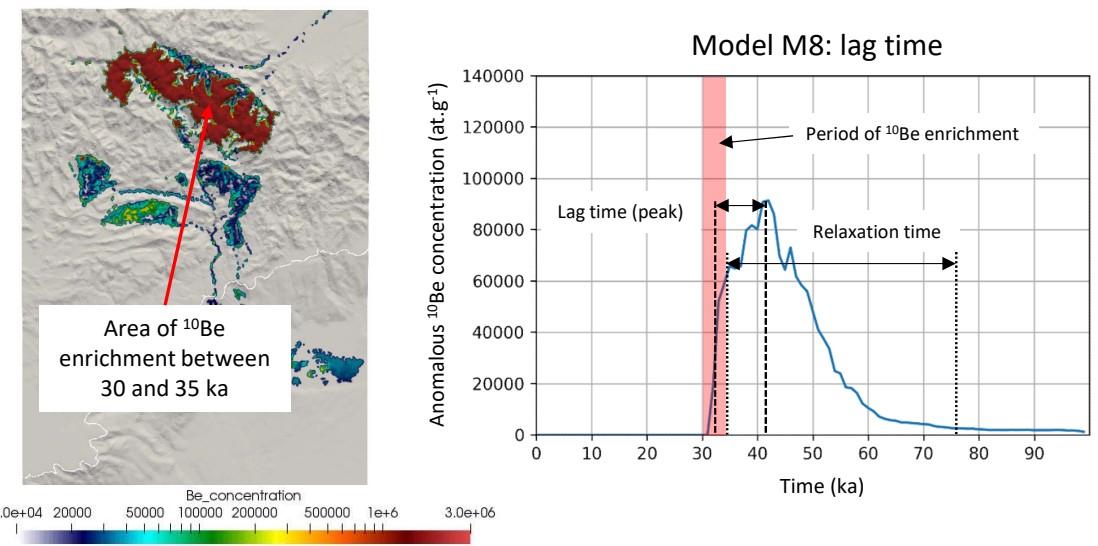

**Figure 7.** Difference in $^{10}$Be$_{MS}$ between two models with the same parameters (Table 3), one of them having an imposed $^{10}$Be-rich ($2x10^6$ at.g$^{-1}$) rocks above 1800 m between 30 and 35 ka (red transparent rectangle). Left panel shows the location of the area of $^{10}$Be enrichment in the Mercantour massif. The time lag (thick dashes) is defined as the interval between the middle of the $^{10}$Be enrichment period in the massif and the age of the largest $^{10}$Be peak in marine sediments. The relaxation time is defined as the interval between the end of the $^{10}$Be enrichment period in the massif and the end of excess $^{10}$Be recorded in marine sediments.

the middle of the $^{10}$Be enrichment period on-land and the largest $^{10}$Be peak recorded in marine sediments is rather large (~10

325   kyr). Moreover, this signal takes a long time to relax: sediments that are richer in $^{10}$Be than the reference simulation still reach the basin ~25 kyr after the end of the enrichment period. These tests suggest that, although the initial time lag is not necessarily very important, the relaxation time, i.e., the time it takes for the $^{10}$Be concentration to return to its background value, can be large. It suggests that, in the case of a succession of climatic events with a frequency greater than this relaxation time, it will therefore be impossible to differentiate between them in the $^{10}$Be signal. We want to stress here that this delay is different from

the time needed to reach cosmogenic radionuclides steady state, as computed in Bierman and Steig (1996). Here, we do not force high $^{10}$Be concentration in the Mercantour massif by reducing drastically the erosion rate - which would indeed take some time for the $^{10}$Be signal to adapt and reach a new steady-state. Instead, we instantaneously apply a large, artificial $^{10}$Be value in the mountain peaks, which should be immediately visible in deposited sediments if the time lag was null.

### 4.2   Reference model for the Var catchment over the last 100 ka

Finally, we try to determine if it is possible to find model parameters for which the result matches both the previously determined river incision rates during the last 30-40 ka in the Nice hinterland (Figure 2) and measured $^{10}$Be$_{MS}$ for the same period

(Mariotti et al., 2021). We chose not to consider the two river gorge sites that have been dated in the upper Tinée valley (Salso Moreno and Isola, see Figure 1), because they likely correspond to very transient post-glacial sediment wash-out that is not possible to model (Rolland et al., 2020). This "Reference" model has been determined by a trial-and-error method, with initial
and boundary conditions similar to the previous series of runs (Figures 8 and 9A, model m9). Sea level variations and ice cover are imposed, and simulations are run for 100 kyr with a time step of 1 kyr. We assume that for the last period of the model, which corresponds to the present-day situation, the precipitation rate should be compatible with current average values (i.e., 700-800mm.yr$^{-1}$, https://meteofrance.com/climat/normales/france/provence-alpes-cote-d-azur/NICE). Here again, flexural isostatic response is computed using a constant effective elastic thickness of 20 km. We also tested different *EET* values
of 10 and 30 km as well as a space-variable *EET*, with larger values on land than at sea. Since variations in *EET* are difficult to constrain and cause only minor differences in the final result, we chose to impose this constant value of 20 km on the entire grid for all models. We first present a model which satisfyingly reproduces measured river incision rates and $^{10}$Be measurements, then we discuss the implications of each parameter (flexure, sea level, ice cover) in the final result. This simulation involves a precipitation rate of 0.3 m.yr$^{-1}$ from 0 to 80 kyr increasing to 0.7 m.yr$^{-1}$ during the last 20 kyr, an initial denudation
rate of 0.2 mm.yr$^{-1}$, a diffusion coefficient of 2.5x10$^{-2}$ m$^2$.yr$^{-1}$ in submarine and river sediments and of 0.1 m$^2$.yr$^{-1}$ in bedrock areas and a constant erodibility coefficient of 3.5x10$^{-6}$ yr$^{-1}$ (Figures 8 and 9A and Table 3). This model satisfyingly reproduces measured incision rates in river channels and yields a slight increase in $^{10}$Be$_{MS}$ in the Var deep sea fan after ~40 ka (Figure 9A). Given the large variability of $^{10}$Be$_{MS}$ values (as depicted by the standard error bars), the modelled $^{10}$Be$_{MS}$ variation is compatible with almost all measurements published in Mariotti et al. (2021). From the simulation outputs, we find
that the increase in $^{10}$Be$_{MS}$ is not due to a change in erosion rate on land: indeed, river incision rates tend to increase after 20 ka due to the release of glacier meltwaters and to increased precipitation rates, while the $^{10}$Be$_{MS}$ also increases twofold. The observed increase in $^{10}$Be$_{MS}$ around 30-40 ka is partly due to the presence of patches of $^{10}$Be-rich sediments deposited in the upper part of the basin after about 30 ka, which slowly feed lower areas of the basin in $^{10}$Be-rich sediments during the run (Figure 8B and 8C). The simulation fit with the measured $^{10}$Be$_{MS}$ may somehow be fortuitous, since the surface concentration
in $^{10}$Be varies at lot, both locally and vertically. This simulation thus shows that: 1) $^{10}$Be concentration at the surface of the submarine basin at a given time can be highly variable; hence, the vertical variation of $^{10}$Be$_{MS}$ can also be variable from one place to another, and 2) it is quite possible to get both increased incision rates on-land and apparent decreased denudation rates in submarine cores, for the same external (climatic) forcing.

Starting from this reference simulation (Figure 9A), we then evaluate the individual effects of lithospheric flexure, ice cover
and sea level changes. A similar simulation without lithospheric flexure predicts slightly lower incision rates for all rivers, with a more dramatic change for the Vésubie River where it drastically decreases (Figure 9B, model M10); meanwhile, both the absolute values of $^{10}$Be$_{MS}$ and their variations are still consistent with measurements. A simulation similar to the reference one but without ice cover (Figure 9C, model M11) predicts slightly too low incision rates for some rivers (Estéron, Lower Tinée), but a good fit to $^{10}$Be$_{MS}$ and to some other rivers (Paillon and Roya), and rather high and constant $^{10}$Be$_{MS}$. Removing sea level
changes does not significantly affect the simulated river incision rates, but gives a lower $^{10}$Be$_{MS}$ signal, with still an increase after -30 ka (Figure 9D, model M12). A simulation with constant precipitation rate gives satisfying results for the rivers incision

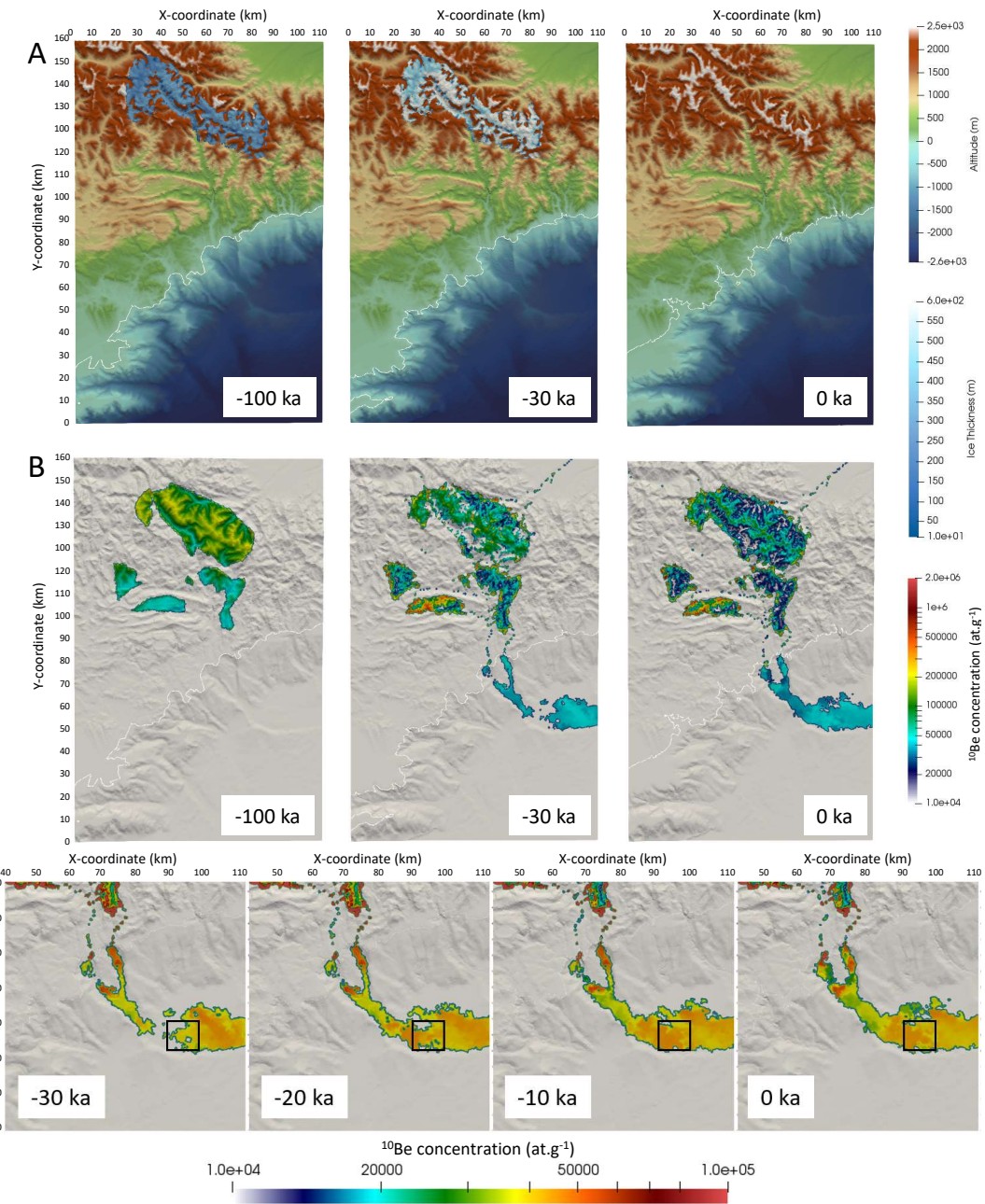

**Figure 8.** Stages of evolution of the reference model from 0 to 100 ka (model time, corresponding to -100 to 0 ka in reality). Upper panels (A) present the topography and ice thickness, middle panels (B) present the surface $^{10}$Be concentration and lower panels (C) present a blow-up of the surface $^{10}$Be concentration in submarine sediments in the last 30 ka, with an adjusted color map. Solid rectangle on the lower panels indicates the area where average $^{10}$Be$_{MS}$ has been computed (see results on Figure 9).

rates but a constant and too low $^{10}Be_{MS}$, especially for the most recent (-20 to 0 ka) period (Figure 9E, model M13). Finally, removing both the flexural isostatic response of the lithosphere and the main sources of load (i.e., ice cover and sea level variations) has large effects both on the river incision rates (especially for the Vésubie River) and on the $^{10}Be_{MS}$ signal (Figures 9F and 9G, models M14 and M15). Interestingly, the last three models without precipitation changes, or without flexure +/- ice cover or sea level changes produce rather stable $^{10}Be_{MS}$ values throughout the model. In fact, the main difference between the reference model M9 and for instance models M13, M14 and M15 is that the first 50 ka of model M9 are characterized by a lower $^{10}Be_{MS}$ concentration. This is possibly due to a relatively lower contribution of high-altitude rocks to the $^{10}Be_{MS}$ signal when the massif is partly covered by ice.

Based on these models, we show that:

- Simulation with constant precipitations or without flexure and without either ice cover or sea level changes do not produce the observed increase in $^{10}Be_{MS}$ after 30-40 ka.

- The large incision rate in the Vésubie River (compared to Bévéra, Roya, Estéron and Paillon) can only be explained by the effect of isostatic rebound, the latter being smaller in the simulations with no glaciers.

- Cosmic ray exposure (CRE) data on river-polished walls are not dense enough to discriminate between constant $(0.5\,\mathrm{mm.yr^{-1}})$ or variable $(0.3$ then $0.7\,\mathrm{mm.yr^{-1}})$ precipitation rates.

- However, a constant precipitation induces more important denudation in the earliest stages of the predicted evolution (compared to the reference simulation where precipitation rate is low), hence producing on average a lower $^{10}Be_{MS}$ at the end of the model.

- A combination of sea level changes, ice cover and lithospheric flexure provides a reasonable fit both to river incision rates and to measured $^{10}Be_{MS}$.

## 5 Discussion and Conclusions

These series of models, where continental sediment deposition is controlled almost exclusively by one parameter (the critical slope for alluvial deposition) show that it is sufficient to induce significant differences in the $^{10}Be$ signal between its production in quartz-bearing rocks of high-altitude massifs and its record in the submarine domain. However, we do not capture here all the details of erosion, transport, sediment mixing and sedimentation, as we do not consider different lithologies, grain sizes, nor any complex mixing law between the various sediment sources. In contrast to the study by Carretier et al. (2020), the lack of grain tracking prevents us from being able to measure particle residence times and confront them with the average geochemical signature of the sediments.

Submarine sediment transport and deposition modelling is almost as simple as on land, and does not consider ocean dynamics (turbulence of turbiditic flows, coastal currents), which are responsible for characteristic sedimentary features such as the Var sedimentary ridge at the bottom of the Var Canyon (Migeon et al., 2001). These phenomena can also contribute to sediment dispersion and smoothing of the $^{10}Be$ signal offshore.

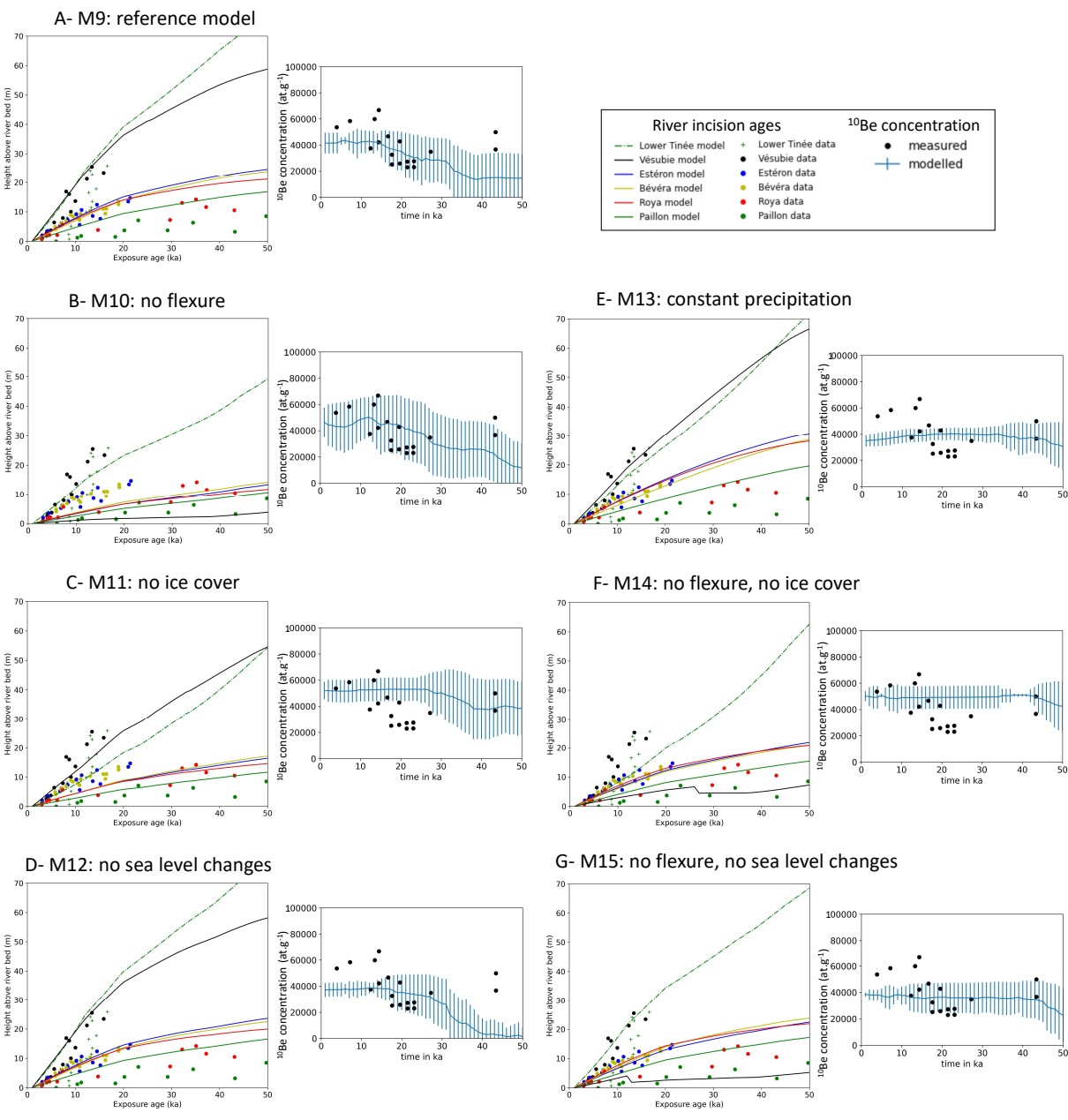

**Figure 9.** Comparison between model outputs and data from CRE ages in river-polished cliffs (left panels, see Figure 2) and $^{10}$Be$_{MS}$ (right panels, after Mariotti et al., 2021). Vertical bars on the right correspond to measured standard deviations in $^{10}$Be$_{MS}$ in the sampled area (Figure 8). See Table 3 for model parameters.

Similarly, it could be interesting to couple our models with glacial erosion models (e.g., Seguinot and Delanay (2021)) in order to better control the variability of glacial erosion and sediment production during the last cycles, and better quantify how it affects [10]Be production and exportation.

    Finally, initial model conditions are another possible source of uncertainty, especially when it comes to our initial topography, which is derived from present-day elevation and therefore limits our ability to run the model over a longer period of times as it

would produce a final topography too different from current one. A better estimate of initial quartz abundances in source rocks (i.e., crystalline lithologies of the Mercantour massif and cenozoic sandstones of the foreland) would also help to reduce the uncertainties associated with quartz and [10]Be transfer from source to sink.

    Our results based on dating and modelling indicate that, while river sands do accurately estimate the average denudation rate of continental catchments for the Var region (provided the latter does not vary at high frequency, i.e., with periods smaller

than the time needed to reach steady-state), it is much less the case for submarine deep-sea sediments. These sediments have a different, and often much smoother signature than continental ones, and record significant lag and/or relaxation times with respect to external forcing, probably due to the geomorphological response of the continental margin both on land and at sea. This area being prone to strong and rapid geomorphological modifications (i.e., transition from narrow bedrock channels to wide, braided rivers) during violent flood events (like during the Alex storm, which took place in October 2020), it could be of

primordial importance to estimate the relaxation time of such events and their role on the long-term landscape evolution and geochemical signature of the sedimentary archives.

    On the long-term, the presence of ice in the massifs where [10]Be production occurs, together with the reworking of alluvial and deltaic sediments during low sea-level periods and vertical motions due to lithospheric flexure largely modify the signal coming from precipitation variations, and can lead to poor estimates of the actual denudation rate variations from [10]Be$_{MS}$. All these

425 effects have been exemplified in this study of the Var catchment where the distance from source to sink is short, precipitation rates are large and the mouth of the main rivers are devoid of any large deltas. Hence, in regions with very large catchments, alluvial plains and deltas, it could be even more difficult to reconstruct past denudation rates from deep sea sediments.

    Our reference simulation highlights the complex interactions between river incision, sea level variations, ice coverage and the resulting isostatic response of the lithosphere. It seems impossible to disentangle the respective role of any of these forcing

specifically, as most of them are interdependent. However, further tests (Figure 9) show that, depending on their location, rivers have a different sensitivity to these parameters: the Estéron, Roya, Paillon and Bévéra are less affected by the parametric changes applied to the reference forcing conditions: their incision rate is only significantly reduced when considering no sea level changes, nor ice cover or flexure (not shown here).

    In contrast, the rivers with the largest incision rates (the Vésubie, and to a lesser extent, the Tinée River) are also the ones

which seem more sensitive to the effect of ice cover and flexural isostatic response of the lithosphere (Figure 9). It is possibly because a significant part of their length (20-40) runs over the Mercantour crystalline massif, i.e. over the area where post-glacial isostatic rebound is important. The sampling sites of these rivers being rather close to their headwaters, they are not sensitive to sea level variations. Quantification of each river sensitivity to local or regional processes like isostatic uplift, sea

level changes, precipitation or ice cover, depending on the dating site location, could therefore be useful to better understand
the respective importance of these external forcing on this South Alpine margin.

*Code and data availability.*   The corresponding version of Badlands with all the data used for this paper can be found here: https://github.com/badlands-model/badlands-Be

*Author contributions.*   CP developped the implementation for Badlands, performed model runs and  wrote the first draft of the paper; TS checked and released the new version of the code and contributed to the paper. YR, VG and LA contributed to the paper.

*Competing interests.*   The authors declare no conflict of interest.

*Acknowledgements.*   This study is part of a project that has been funded by the French Geological Survey (Bureau de Recherches Géologiques et Minières; BRGM) through the national program "Référentiel Géologique de France" (RGF-Alpes). This work has been supported by the French government, through the UCA-JEDI Investments in the Future project managed by the National Research Agency (ANR) with the reference number ANR-15-IDEX-01. Fruitful discussions with Guillaume Duclaux (Geoazur) were greatly appreciated. The authors thank
Sebastien Carretier and Yanyan Wang for their insighful comments on the first version of this manuscript, and to Associate Editor Simon Mudd for handling the review process and for the final review of this paper.

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
