# Peer review of "River incision, 10Be production and transport in a source-to-sink sediment system (Var catchment, SW Alps)"

_EGUsphere, 2022_

## Author Response (AR1)

Dear Editor,

Here are the detailed responses to both referees' comments on our manuscript submitted to Earth Surface Dynamics (ESurf). We have uploaded a revised version with marked changes as follows: referee #1 in red, referee #2 in dark blue and other revisions in cyan.
A bug was fixed in the code (under certain circumstances, some quartz was lost during the deposition in alluvial plains). Although it does not change much the model results, all the models have been re-run with the new version of the code. We have modified the figures in order to make them easier to read and understand, following the referees' remarks. The line numbers refer to the version of the manuscript with marked changes.

Sincerely yours,
Carole Petit, on the behalf of all co-authors.

Referee #1

Description of the model and parameter values
The interest of the model is that it couples surface, terrestrial and submarine processes, isostasy by flexure, and the evolution of the cosmogenic isotope concentration. This implies a choice of processes, and a consequent number of parameters. These choices are not sufficiently explained (at least a table with all parameter values is needed).

*R1: The parameters that were used to run the "best-fit" model are in the Badlands github repository, which link is given at the end of the paper. However, we acknowledge that a table with all parameters values for the presented models is required in the paper. It is now presented in the revised version (Table 3).*

- The river erosion law is classical but the values of the parameters K, m and n are not all indicated nor justified, whereas it is known that the response of the rivers strongly depends on these values (especially n).

*R2: Indeed, we used the "classical" m and n exponent values of the stream power law and did not perform a systematic analysis of these parameters on the different catchments of the study area. The m and n exponents and the erodibility coefficient K are difficult to constrain independently, and can change with time (K) or from a channel to another (m and n) but at this scale, we cannot take into account this variability.*
*Chi-plot analysis in the Vesubie river has revealed values close to these ones (Saillard et al., 2014 and Petit et al., 2017) so we believe they are a satisfying approximation. We tuned the value of K with fixed m and n until we obtained consistent incision and denudation rates over the catchment (compared to existing data).*
*This is now explained in the revised version of the manuscript (lines 134-136).*

- The slope erosion law is a simple diffusion law whereas it is stated that it describes linear or non-linear diffusion. The non-linear component, which would indeed be better suited to describe mass wasting on steep slopes, does not appear in equation (3). The values of Kd are not justified in the paper, either for bedrock or sediment (is there a distinction?).

*R3: In Badlands, one can choose between two options to simulate these processes. In the first, we use a linear diffusion law commonly referred to as soil creep (the one presented in the manuscript eq. 3). In addition, we define 2 diffusion coefficients: one for the marine environment, the other for the continental part. An alternative formulation is also available and expressed as a non-linear formulation of diffusive hillslope transport, assuming that flux rates increase to infinity if slope values approach a critical slope $S_c$. The Kd value is set to 0 on land in test models in order to minimise the effects of diffusive processes on the $^{10}$Be signal. It is also rather low in the submarine area because otherwise, the smoothing effect on the $^{10}$Be signal would be due to submarine diffusion rather than to other processes (like sediment storage). Still we could not completely set it to zero because we need to insure a realistic deposits geometry. This is now explained in the text (lines 140-144).*

- Sedimentation occurs below imposed thresholds, on land and at sea, but these thresholds are neither justified nor quantified in the manuscript. However, the dynamics of rivers depend in reality on the cover and tool effects of sediment. How are these effects treated here? It seems that sediment can be re-eroded: in this case, how do they erode? With the same K, Kd, m, n parameters as for the bedrock? (…) A better description of the sediment layer behaviour (deposition, sediment recycling), onshore and offshore sediment dynamics would help interpret the model results.

*R4: For the continental erosion, there are indeed different formulations available in Badlands from detachment-limited to transport-limited (https://badlands.readthedocs.io/en/latest/xml.html#transport-limited-processes). Here, the detachment-limited approximation is used so the tool and cover approach suggested by the reviewer is not accounted for. Yet even when applying the SPL, the approach allows to deposit sediment under a given slope threshold (value of 0.005 in the XmL file) and in depression areas.*
*For the marine part, the formulation is based on Petit et al. (2015) and Thran et al. (2020) for hyperpycnal flows and otherwise rely on the diffusion law as explained in the text. Sediment deposition and erosion obey basically the same laws on land and at sea (except for the diffusive part), with some specific parameters related to the marine domain (like a lower "effective" slope for instance). We want to stress that the objective here is not to model precisely offshore sedimentation dynamics, but rather to be able to transfer sediments from the coastal area to the deep submarine basin, in order to get close to the actual situation of the Var system. Most LEM consider immediate sediment deposition close to the coastline, this is why this modification was implemented to Badlands in the two above-cited papers.*

For example, the authors interpret the bad record of continental erosion by offshore detrital 10Be concentration as the result of sediment recycling in the land rivers. This interpretation may be tested by running a simulation without sedimentation on land and verifying that the offshore detrital signal is different.

*R5: It is, in fact, due both to sediment storage on land and at sea in the submarine delta and canyons. While it is difficult to tune off completely alluvial plain deposition only on land, we add some tests which show that if we strongly limit sediment storage on land and force deposition in the coastal area, we obtain more consistent records in the upper part of the delta, while in the lower part the signal is delayed (due to progradation). This is now*

*explained in the revised version (lines 145-147 + 2 new models on Figure 6 with explanations lines 264-267).*

- The model seems to calculate the sediment flux Qs using an empirical law (equation 1). Why do we need an empirical law to calculate Qs when it should be derived from the difference between erosion and sedimentation upstream of each pixel? Is this empirical law used to distinguish between fine (which is what it was designed for) and coarse sediments on which 10Be is measured? Is it verified in the model that the flux calculated with equation (1) is lower or equal to the one corresponding to the sum of the net erosion and sedimentation upstream?

*R6: This empirical law is related to the suspended load component and not the total sediment flux, which actually work based on upstream erosion/sedimentation as described by the reviewer. While it can be critical to use one or the other approaches if we aim at triggering hyperpycnal flows for a given threshold density, here it has no impact as we force hyperpycnal deposits to occur all the time (threshold flow density is lower than water density). See modifications in the text lines 155-157.*

- The parameters of the flexural model should also be given.

*R7: They are now included in the text (lines 166-172).*

- The 'best-fit' model must be better characterized (method and parameter values), as well as the method used to determine the parameters of this model (Monte Carlo approach with minimisation of a misfit function, for example). Given the number of parameters to be adjusted, I assume that choices have been made to keep some constant while varying others. Details of this procedure are needed.

*R8: It is always complicated to define objectively what is the best-fit model, given the range of parameters tested and the available data: first, channel points where river incision rates are close to the actual ones, but not exactly on the same spot (due to model resolution). Second, the area where marine sediments are sampled in the model is wide because we want to take into account the lateral variability of the $^{10}$Be signal, but in nature it has been sampled on a single spot along a vertical core. For these reasons, a misfit function would not provide more information than a simple visual inspection of the model outputs.*
*Instead of a "best-fit" model, we modified the text in order to present what we call a "reference" model, i.e. a model in which we obtain a reasonable fit to observations, while considering some parameters (sea level, ice cover, effective elastic thickness) as fixed.*
*It is true that some parameters were not allow to vary a lot (for instance, precipitation rates were allowed to vary between 0.2 and 2 m.yr$^{-1}$ and m and n were fixed). In the last series of models, we tuned the K coefficient while assuming a recent precipitation rate of 0.7 m.yr$^{-1}$ which is close to the actual mean annual precipitation rate in this area. EET values of 10 to 30 km were tested, as well as a grid of space-variable EET but the results were not significantly affected by these variations. This is now explained in the text (modifications lines 352-360).*

- An important result is the fact that onshore sediments give a good representation of erosion, while offshore sediments may have a cosmogenic signal de-correlated from erosion.

What is happening on the offshore slope that changes the signal? Is it explained by sediment recycling from the offshore fan? Is this result the same if we sample the offshore fan apex?

*R9: Some answers to these questions are provided by the new tests presented Figure 6. Sediment recycling occurs both on land and at sea. Lines 264 to 267.*

The calculation of the cosmogenic isotope concentration is based on the differential equation (6) which is a simplification of the general equation involving the concentration gradient with depth. This simplification allows the evolution of the concentration to be written as a function of the surface concentration (Eq. 6 of Knudsen et al. 2019). However, I confess that I did not understand how equation (7) was established... It seems to come from Knudsen et al. (2019) but I did not manage to find this equation in their article. If this equation is correct, I think it is important to demonstrate it and explain why it has an advantage over a simple numerical solution of equation (6) (where $dN/dt=(…)$ could simply be approximated by $[N(t+dt)-N(t)]/dt=(...)$ so that $N(t+dt) = N(t) \, dt + (...)dt$ could be computed on each pixel at each time scale). I think it would also be useful to explain how one can avoid from calculating a depth concentration profile on each pixel at each time step since at a time $t+dt$, the surface concentration takes into account the concentration that was previously at a depth (Epsilon x dt). Different previous approaches based on grain tracking may be worth to cite (Repka et al., 1997; Codilean et al., 2010). Similarly, a comparison with the approach of Yanites et al. (2009) may be useful.

*R10: First, there is a typo mistake in this equation (minus sign before the exponent in the right member), so we apologize for it... Second, it is just the same equation as eqn 7 in Knudsen et al. (2019) but with the exponent term factorized. We wrote it like that because it is the way it appears in the code, but we acknowledge that we'd better stick to the original equation. It is now corrected.*
*The advantage of this Eulerian formulation as described in Knudsen et al. (2019) is that we can compute a TCN concentration at the surface without having to define vertical concentration profiles at each point, which would be time consuming. The velocity of vertical rock advection is the erosion rate and this formulation takes into account the vertical variation of TCN concentration. We tried to better explain this point in the revised version of the paper (lines 215-217).*
*We have also added the references about grain tracking approaches in the text (lines 56-64 and 407-410.*

It is not clear to me how to calculate the 10Be concentration in the sediment on each pixel. It is stated that the 10Be concentration of the deposited sediments is a kind of average weighted by the eroded volumes upstream, but how do we know the origin and composition of the locally deposited sediment? Is the 10Be concentration of the sediments transported from one cell to another tracked from upstream to downstream?

*R11: yes, we can track how much sediment are flowing through each cell, and how much is deposited and each time step. Badlands also allows one to track the relative importance of different sources (not used here). We simply add two other variables describing the sediment flux: the volume of quartz and the number of $^{10}Be$ atoms it contains.*

Absence of a "Discussion" section

The choice of simplification of the processes involved and the parameters used requires a discussion on the robustness of the results with respect to other choices. For example, how does the choice of sedimentation threshold value change the result? Are the results dependent on m and n? Does the absence of attrition or different grain sizes have an implication on the results? Variation in river width is not taken into account: is this a problem?

*R12: We agree that the results are not sufficiently described, especially with respect to model limitations.*

*As explained above, changing m and n values also requires to change K (in order to still be able to fit measured incision rates) so we think we cannot do much better than choosing arbitrarily m=0.5 and n=1, and then tune K. The sediment threshold (here, the critical slope for alluvial plain deposition) is definitely important but it is already low (0.5% slope), so our model does not really favour sediment storage. Finally, we cannot really estimate the behaviour of different grain sizes in Badlands. In Mariotti's papers, [10]Be measurements were performed on two different grain populations and have shown that there is indeed some variability in [10]Be concentration depending on grain sizes. Our models show that there can be also some lateral variability in the concentration, depending on how the sediments are transported in the submarine domain (new Figure 6).*

*Variation in river width (with channel length) is already taken into account by the m<1 exponent in the SPL, otherwise it should be equal to 1 (e.g., Whipple and Tucker, 1999; Finnegan et al., 2005). Other variations are not taken into account.*

*The revised version now discusses these points (lines 403-422), and there is now a discussion-conclusion section.*

Specific comments

Line 79 How were a and b estimated? With what data? On what grain size?

*R13: a and b values were directly taken from Syvistky et al. (2000) (from measurements in the Var river) and rounded to 1.e-3 and 1.6, respectively. There is no grain size dependency. See text lines 90-91.*

L 103-105 This is not inconsistent. The incisions are local compared to the average erosion estimated by 10Be in detrital river sediment.

*R14: It is not necessarily inconsistent as such, but the interpretations are fundamentally different. While river incision rates have been interpreted as reflecting more intense erosion during the last interglacial, geochemical analyses of submarine cores have been interpreted oppositely (i.e., larger erosion during glacial periods). See text lines 117-123.*

L117 Define and explain how this threshold is chosen (from what?) and the choice of its value

*R15: Its value is of 0.5% (0.005). We have chosen this low value because we did not want to favour alluvial plain deposition. Lines 145-146*

L 130 Is this very simple model able to reproduce the main characteristics of the alluvial dynamics (avulsion, recycling, etc) which seem to be determinant for understanding how the 10Be signal is modified on land and in the sea?

*R16: this model allows us to account for sediment erosion, transport and deposition in the sea with almost the same hypotheses and limitations and on land (with the differences explained in the text). So it can reproduce exactly the same things. What it cannot do is take into account all that is related to ocean dynamics (turbulence, submarine currents, and so on…). See the new discussion-conclusion section.*

L147 Could you justify that glacial erosion is proportional to the slope?

*R17: glacial erosion depends on the shear traction exerted by the glacier on the bedrock (see Boulton, 1996) so we consider it varies with the thickness of the glacial tongue (ice load) and on the bedrock slope (ice velocity). This is now explained in the text lines 188-190.*

L178 How is this calibration done in practice? Is it done by a Monte-Carlo type approach? Is there only one solution (one set of parameters)?  Please, give a table with the parameter values.

*R18: Trial and error method. A table with the parameters is now provided (Table 3).*

L180 What do you mean by "evaluated"?

*R19: Computed. This has been corrected.*

L183 The sample Ves161-1 is also very different from the model result. Do you have an explanation for these deviations from the model?

*R20: Ves16_1 has a $^{10}$Be concentration that is only slightly larger than 1/1 curve, but because it has a very large production, rate the resulting denudation rate appears much lower than the 1/1 curve. So in this case, only a slight overestimation of the $^{10}$Be concentration can lead to a large underestimation of the denudation rate. The text has been modified accordingly (lines 232-235).*

L190 Why is it necessary to use an empirical relationship to calculate the sediment load when it can be derived in the model from the net erosion/sedimentation calculated upstream?

*R21: see R6*

L191 What is the value of this threshold?

*R22: 0.5%. This is now in the parameter table and we have tested other values for this threshold.*

Equation (9) Q is already used for water flow.

*R23: Corrected.*

L199 Equation 7 is not a solution of the general equation with stationary erosion rate. Do you mean equation 8?

*R24: Yes, this has been corrected.*

L209-210 Since the method is precisely based on the ratio of quartz fluxes to 10Be fluxes, there should be no overestimation. Unless I have misunderstood: do you mean that the concentration of sediments from glacial erosion are under-concentrated because the ice protected them from cosmic radiation?

**The following minor points have been corrected directly in the text and/or figures.**

*L 216 Could you clarify what the "in-situ" and "actual denudation rate" are?*

*L220 What are the analytical solutions?*

*L235 Is it possible to test this hypothesis by preventing the sediment from settling on land (with a different threshold for deposition)? I think this would be convincing.*

*L239 Source of what?*

*L241 But the effect of glacier cover is well known. In general it is taken into account when calculating erosion rates.*

*L256-259 This requires a Figure*

*Figure 7: where is the second model? I see only one blue curve...*

*L278 This is an interesting point*

*Figure 9 Where is the best-fit model? Is it the reference one? The text is too small in these panels. I suggest to split it into different figures or increase the text size and reorganize the panels.*

Referee #2 (modifications in dark blue in the revised manuscript)

**1 Major questions:**

1- The calibration model that is presented in section 3.3. I wonder how representative the smoothed topography from DEM is. For example, how much the high peaks in the massif area are smoothed. This is particularly important because these are likely the major contributors to Be10. If it is a matter of computation efficiency to decrease the grid resolution, did the authors try with some of the small catchments, for example, the Tinee

river or the Vesubie river, to evaluate the smoothed topography and see if the same model parameters calibrated from the whole Var catchment can produce the same results in these small catchments?

*R1- The use of a smoothed tomography is not a real problem as long as the shape of the drainage network is preserved. It is not true that the high peaks are the most important contributors to $^{10}$Be: indeed, $^{10}$Be production is high because of the altitude, but the erosion rates are very low. The major contributors to $^{10}$Be sediments are average-altitude areas where the 10Be production is important and fluvial incision starts to become also important (large slope and increasing drainage area). We mention the possible impact of a low-resolution topography in the Discussion section (lines 418-420).*

A second question is what is the uplift rate imposed in the model? Is it zero uplift? I didn't find any clue about it. The authors should clarify it and give reasons.

*R2- No uplift is imposed. Tectonic uplift rates are not well constrained and must be very slow (0.1 or 0.2 mm.yr$^{-1}$). Lines 172-173.*

Flexural uplift is explored later in other models (section 3.4 and 4), given a 20 km of EET is used, the wavelength of flexural uplift is longer than the Var catchment size, that is saying, perhaps the spatial variations of flexural uplift is minimal. So if zero-uplift is used, why bother to test the flexural uplift later? Please give reasons, for example, is some area uplifted faster than the other regions due to the flexural uplift, and therefore a change of topographic slope (this is important for sediment transport) is expected? The flexural uplift is never tested alone when keeping the other factors constant in models of Figure 6. The authors could either provide arguments about the necessity of imposing uplift in the model, or make an additional model test of flexural uplift only to see what the model tells us.

*R3: In Badlands we can apply different types of kinematic conditions: i) "tectonic"-related motions due to fault activity for instance, and ii) vertical motions due to flexural response. The first ones are imposed from the beginning of the model, the second ones are computed from the load variation (due to erosion, sedimentation, sea level changes) during the model run. There is no incompatibility in setting zero tectonic motion while allowing for flexural isostasy to occur. Flexural uplift is the response of the lithosphere to loading or unloading (during the model), and since most of the loading/unloading is due to ice melting and sea level changes, it makes no sense to test flexure alone. Still, we have added two models (M14 and M15) where we test both the absence of flexure and the absence of the main sources of lithospheric loading (ice and water). This is discussed in the text lines 385-389.*

(2) I wish the authors can clarify how the inferred denudation rate from the marine sediment and the in-situ sediment is calculated in section 3.4 and 4. Especially, whether or not the inferred erosion rate is corrected from ice-cover and quartz-fertility or not. In practice, the calculation of a basin-averaged erosion rate from a cosmogenic sample should make these corrections when they vary dramatically in the catchment. And this shouldn't cause any discrepancy from model data, at least not the discrepancy between the erosion rate and the inferred-rate from in-situ sediment (the blue dots and red line in Figure 6A). Please correct me if I miss some processes.

*R4: The computation was carried on taking into account quartz fertility, but the production rate was averaged only on areas where production is > 0 so it did not take into account the ice cover. This is now corrected (new figure 6). Lines 207-210.*

Figure 6A, I see an acceptable match between the marine sediment, the "real erosion rate" and in-situ sediment from 85-100 ka, so what does it mean considering the absence of any variations of precipitation, ice cover, etc.?

*R5: Depending on the time period considered and on the area where submarine sediments are collected, and considering that the average 10Be concentration in submarine sediments only displays smooth variations with time, you can find time periods during which the three values match and roughly correspond either to the maximum or to the minimum erosion rate.*

(3) The so-called lag time that is observed in the models in Figure 6. In line 220-223, the authors argued there is an apparent lag time for an abrupt increase in precipitation. Actually, from Figure 6A, the lag time is more or less the same as well for an abrupt drop in precipitation rate. It is not "less visible" as the authors argued in the text. I wonder whether it is a numerical error because the production rate exponentially decays in a depth profile, but the erosion rate more or less linearly changes with the precipitation rate (to the power of m in Equation 2). (A little distracted from the main point, but the authors should put the term of precipitation in Equation 2 for the purpose of this paper.) If the timestep is significantly small, would the authors still expect the same lag time in the model scenarios of Figure 6? This might propagate into the lag time (10-15 ka) of the marine sediment when the ice cover significantly affected the Be10 production area (Figure 6C, D). I wonder what the authors think about this problem.

*R6: This is a good question, but in fact the two things are completely different. In Fig 6A it takes some time indeed to the red curve to reach a peak (on the new figure it is well visible at around 78 ka), while the change in precipitation rate is instantaneous. This is because it takes some time to the average $^{10}$Be concentration in sediments to reach their steady-state value (with respect to the denudation rate). In figure 7, the $^{10}$Be signal itself is instantaneous, so there is no steady-state value to reach. If there was no lag time, it would take only one timestep (1000 yrs) to reach the deep basin. Lines 342-346.*

(4) Residence time of sediment in the system. Following question (3), if we assume the time lag due to numerical errors is quantifiable or ignorable, why the offshore marine sediment showed a constant value after 40 ka regardless of the fluctuated erosion rate in Figure 6A? I think the sediment residence time is underexplored in section 4. Since sediment is trackable (?) in Badland, why not track the sediment thickness, sediment patches, and the number of nuclides carried in sediment throughout the modeled domain? A temporal inventory of sediment and nuclide (eroded material, sediment deposited in the Var basin, and in the offshore channel and depocenter, and the nuclide carried within the sediment) will clearly reflect what is missing in the offshore sediment. The authors might have tried in this direction because the model they presented in Figure 7 was also trying to understand the lag time by imposing a slow erosion rate (and therefore high Be10 concentration) on a certain part of the basin. But the models in Figure 7 were presented very briefly in the text which from my perspective, should rather be expanded in the text. Models presented in Figure 7,

what is the substantial difference between this model and a model of applying the ice cover only (no sea level fluctuation, no flexure, no precipitation temporal variation)? Figure 6 also missed a model showing the net effect of ice cover only. If the sediment storage is not explored, it is really hard to understand the text line 277-279 "The observed increase in 10BeMS around 40 ka is due to the presence of a patch of 10Be-rich sediments deposited in the basin in the early stages of the model, which slowly migrates towards the lowest areas of the basin during the run" because it was not presented in Figure 8, or Figure 9 and this is a very necessary proof for the authors to make a conclusion here.

*R7: It is true that we can track a lot of things in Badlands (sediment thickness, different rock sources, age of deposition, nuclide concentrations) but we cannot record, for a given sedimentary deposit, the exact location of all its sources. This is almost impossible, as it would require to store a huge amount of data for a given sediment node. Hence the idea to impose an artificial $^{10}$Be peak. We remind here that we DO NOT impose a low denudation rate (which would indeed give a high $^{10}$Be concentration, but after a certain time). We use $^{10}$Be as a marker of the sediment source (in the Mercantour massif) by imposing a large $^{10}$Be concentration in the bedrock for a short period of time, and then trace the product of erosion of this source in the submarine basin. It is quite different (see R6). The purpose here is not to test how the lag time varies with ice load, sea level and so on, but just to show that with the simplest setup, we already observe a significant difference between the source signal and the sink record.*

**2 Some suggestions on figures:**

(1) Figure 2. Since the authors already named the 9 sites with numbers in Figure 1, I would suggest the authors put the site numbers together with the site names in the key (at the top right of the figure). So Line 91- 94 can be phrased to refer to either the site numbers, or the site names, and causes no confusion.

*OK (Figure 2 has been modified accordingly)*

(2) I suggest the authors put the outline of the Var catchment and the submarine drilling cores (B2017) in Figure 3a, Figure 5, and Figure 8 for a better spatial reference, similar to Figure 1.

*OK*

(3) The colorbar of Be concentration in Figure 5, and Figure 8 (the three sub-figures in the second row) doesn't show the spatial variation well. I suggest the authors adjust the colorbar so that the variations within the sediment patches are shown with more distinguishable colors. I see some spatial variations of concentration in the marine sediment in Figure 5. Is it showing one timestep or the integral of all steps? The authors could clarify it in the captions and explain why there is a spatial variation if it is showing one timestep.

*OK. There is now a blow-up on the submarine domain with an adapted color map.*

**3 Suggestions for presenting models parameters:**

(1) A table of parameters and values for each model run will be necessary for understanding the model results. For example, in section 3.3 (Line 179), what are the "initial parameters" that were calibrated, and what are the values used for each parameter? Important parameters of models presented in Figure 4, Figure 5, Figure 6, and Figure 8 should all be listed in a table, either in the main text or goes to the supplements. It might be helpful if the authors can refer to each model run, for example, model 1, model 2, model 3,… consistently throughout the main text and the tables.

*This remark was also made by referee #1. This is now corrected.*

(2) Precipitation is missing from Equation (2).

*OK, corrected. Lines 128-129.*

**4 Suggestions on the paper structure:**

A discussion section is definitely needed to discuss the results. The discussions can focus on some of the topics that I put in the **Major questions**.

**5 A few line-by-line comments:**

***R: The following minor comments have been taken into account and the text and/or figures have been corrected accordingly***

Line 7: "nearly" = "nearby"

Line 79: The estimated value of sediment flux is missing.

Line 87: $^{10}Be_{MS}$, what does it mean?

Line 91: "red-brown dots on 2", should be "red-brown dots on Figure 2"

Line 93: "3, 4, 5, 6,8, orange, yellow and green dots on 2", should be "site 3,4,5,6 and 8 on Figure 2".

Line 94: "… blue dots on 2", the same problem as Line 93.

Figure 2: Since the authors already named the 9 sites with numbers in Figure 1, I would suggest the authors put the site numbers together with the site names in the key (at the top right of the figure). So Line 91- 94 can be phrased to refer to either the site numbers, or the site names, and causes no confusion.

Line 161: give refs here.

Line 163-164: symbols are missing.

Line 179: which parameters? Need to present clearly.

Equation (8): how the initial long-term erosion rate is determined? Is the long-term erosion rate everywhere the same for all catchments?

Line 200: $^{10}Be_{MS}$, is it Be10 concentration of marine sediment? I see this symbol is used throughout the paper till the end, please give the exact meaning of this symbol in the first place where it appears.

Line 202: (as shown in 5)  (as shown in Fig.5)

Line 203: "use the same equation…denudation rate variations". Figure 5 presented the denudation rate, so please delete the "variations".

Line 226: delete "specific"

Line 324: (9)  (Fig. 9). This is not the last place of the same problem as found in Line 202. Almost every figure was referred to in this way. Please correct the same problem throughout the paper.

---

## Referee Report (RR1)

**Review of the revised manuscript "River incision, [10]Be production and transport in a source-to-sink sediment system (Var catchment, SW Alps)"**

Dr. Yanyan Wang

The manuscript is greatly improved in the presentation of the models, model variables, and results. This manuscript clearly demonstrated the potential pitfalls of interpreting the inferred erosion rate from cosmogenic nuclide concentration of marine sediment. I recommend accept this manuscript  or with some very minor revisions which I listed below.

A minor general point about the terminology of different erosion rates. Throughout the paper, there are four kinds of erosion rate, that are, 1) erosion rate inferred from measured cosmogenic concentration either of river sediment or marine sediment (the data presented in Figure 2); 2) direct LEM model results (line 256); 3) the modeled cosmogenic concentration-inferred erosion rate assuming steady state erosion, i.e., Eqn. 10 and Eqn. 11 for "in-situ" sediment (from the concentration of $N_{ex}$); and, for 4) marine sediment (from the concentration of the "$10Be_{MS}$"). In the text, I can see the authors trying to use "actual" or "real" to distinguish them but these two words can be ambiguous. Following the authors' narrative, one way I can think about is model-$10Be_{MS}$-inferred, direct model results, or measured $10Be_{MS}$-inferred to distinguish the measured concentration against modeled concentration. The authors can also stick to the simple ones like "in-situ erosion rate", or "deep basin"/ "canyon"-inferred erosion rate as the authors used these terms in Figure 6 top right panel. To do so, one or two additional clarifying sentences inserted into somewhere around line 255 to Line 257 will be proper. And make sure the terminology is used consistently throughout the paper.

Otherwise, the three places below should be corrected.

Figure 3 captions, first line: "contour"→ "outline"

Line 216: in between "surface" and "is", misses a comma

Line 252: "such as" → "such that"

---

## Author Response (AR2)

**Revisions (S. Mudd, associate editor)**

Dear Simon Mudd,

Thank you for your review of our manuscript. Here is how we took into account your comments. We have uploaded a revised version of the text with marked changes in blue, and a clean version without marked changes.

Sincerely yours,

Carole Petit.

Minor edits (English language) have been corrected directly in the text

Page 3: "real" erosion rate: we have replaced this word by "simulated"

Page 5: the units of a and b have no physical meaning but their values have been determined for $C_s$ in kg.m$^{-3}$ and $Q_w$ in m$^3$.s$^{-1}$. Quoting Syvitski et al., "the rating coefficient has variable units and depends on the value of the rating exponent b: $[M/L^3][T/L^3]^b$". This is now explained in the text.

Page 7: the value of *m/n* has been taken from previous studies (like in Saillard et al., 2014) so we infer from it m=0.5 and n=1 as is often done. Now indicated in the text.

Page 7 and 8: About sediment flux and deposition: From the detachment-limited law (Eq. 2), sediment entrainment rates are obtained on every node in the landscape from upstream to downstream regions and local sediment flux moving out of a cell equals the flux of sediment flowing in plus the local erosion rate. Because we use a detachment-limited approach, no restriction is applied to the sediment concentration a given river is able to carry, and rivers with limited transport capacity are not considered in this study. As rivers flow across the landscape, sediment deposition might occur under three circumstances: (1) either when the channel slope falls below a given threshold (alluvial plain deposition, see Table 3), (2) when the rivers reach their baselevel or (3) when entering a depression (endorheic basins). In these cases, available sediment fluxes carried by rivers are used to compute the volume of sediments to deposit. If transported sediment fluxes, when deposited, are insufficient to fill the depression or to reach the prescribed channel slope threshold, all sediments would be deposited and the outgoing river sediment concentration would be null. If, on the other hand, the available sediment flux exceeds the required deposition volume, the excess flux will be carried out to the downstream nodes. We briefly recall this in the revised version of the paper.

Page 9: The flux of sediments coming from glacial erosion during glaciations is very low because glacial erosion is simulated by local processes (diffusion) and river discharge is set to zero beneath glaciers. As stated in the text, this probably underestimates the role of reworked glacial sediments in the observed [10]Be signal. Still, it allows us to take into account surface denudation by glaciers during glaciation periods, that leads to lower-than-average [10]Be concentration in glaciated areas.

Page 11: about the comparison with Mariotti's study and ours: we do not apply a constant erosion rate to every pixel, as they do (no particular erosive process is considered when computing denudation rates from the average [10]Be concentration in river sands). Instead, we apply a constant precipitation rate and simulate hillslope creep, water discharge and subsequent river incision (and [10]Be production, transport and deposition). Therefore, some pixels erode faster than others depending on their location (river channel, hillslopes, altitude). This explains why our simulated [10]Be concentration is slightly different from theirs.

Page 13: the production rate is also slightly different because the topography is smoother and the map of quartz-bearing rocks is a bit simpler than the one used in Mariotti et al., 2019 (due to the surface grid mesh). If, for instance, we have quartz-bearing rocks in lower altitude areas than in Mariotti et al., we will find a lower average production rate in the given catchment.

Page 20: about sediment storage: it is true that the initial topography contains low topographic gradient areas where sediment storage can occur, and which can be progressively dissected later on. However, this occurs at the very early stages (2-3 time steps, i.e. 2 to 3 ka) of the model, while the "patch" of $^{10}$Be-rich sediments which is observed on model M9 in the Var delta and upper submarine canyon starts to develop much later (after 30-40 ka), so we believe it is not due to initial conditions. In the reference model, the final $^{10}$Be concentration (-30 to 0 ka in "real" time) in marine sediments is not larger than in the other models (it is even a bit lower than in some models). The difference between this model and, for instance, models M13, M14 and M15 is that the earliest stages of model M9 (-100 to -30 ka) are characterized by a lower $^{10}$Be concentration in submarine sediments. It is not easy to determine precisely what is the relative contribution of the different sources to the final $^{10}$Be concentration of sedimentary deposits. However, from what we could infer from the reference model, up to -50 ka there is little contribution of submarine sediments to the deep $^{10}$Be record. After that time, sediments from the upper var canyon start to become reworked and contribute to $^{10}$Be enrichment in the sediments of the deep basin.